# AhaTrans: A Hierarchical Adaptive Transfer Learning Framework for Cross-City Traffic Flow Prediction

## Abstract

Accurate prediction of urban traffic flow is essential for optimizing traffic management, enhancing urban planning, and promoting the development of smart cities. Due to the difficulty of data acquisition in many cities, data scarcity arises, significantly impeding the practical application of deep learning techniques. Consequently, researchers have turned to transfer learning for mitigating data scarcity through cross-city knowledge interaction. However, existing transfer learning methods lack precision and discrimination in spatio-temporal feature extraction, thereby restricting the predictive performance. Moreover, these approaches frequently fail to adequately account for the disparities between the source and target cities, resulting in the loss of essential knowledge and, at times, the introduction of detrimental knowledge into the target city. To overcome these challenges, we novelly introduce **A h**ierarchical **a**daptive **Trans**fer Learning Framework (**AhaTrans**), which ensures precise feature learning as well as effective, non-detrimental knowledge transfer in cross-city traffic flow prediction by focusing on three key levels: model architecture, feature representation, and data adaptation. Specifically, AhaTrans consists of the following three core modules: i) Guarded Transfer Experts Network (GTEN), which clearly distinguishes between shared and city-specific experts, enabling the target city to access beneficial knowledge from the source city while preventing harmful knowledge; ii) Spatial-Temporal Contrastive Embedding Module (STCE), which enhances the representation of spatio-temporal features through contrastive learning; iii) Transfer-Based Reweighting Module (TBR), which dynamically adjusts source city samples to extract knowledge most relevant for the target city's traffic patterns. Extensive experiments demonstrate that AhaTrans significantly outperforms existing methods, substantially improving the accuracy of traffic flow prediction while exhibiting excellent robustness and generalization capabilities.

## 1 Introduction

Urban traffic flow prediction plays a vital role in the development of smart cities and the optimization of intelligent traffic systems (Zheng, 2015; Mazimpaka & Timpf, 2016; Yuan et al., 2020). It constitutes a typical spatio-temporal prediction task, aimed at forecasting future patterns through the analysis of historical traffic data (Fang et al., 2022). Modern urban areas generate spatio-temporal data via GPS, mobile devices, and remote sensing technologies (Gonzalez et al., 2008; Zheng et al., 2008; Weng, 2012). These data sources are diverse and multimodal, encompassing trajectories of bikes and taxis, along with public transit usage. Traditional statistical and regression models (Zhang, 2003; Lippi et al., 2013) often face challenges when dealing with these complex and correlated datasets. Hence, effectively understanding and utilizing these data is crucial for accurate predictions. Recently, deep learning methods have shown remarkable performance in traffic flow prediction. Researchers have utilized increasingly sophisticated networks, such as Convolutional Neural Networks, Recurrent Neural Networks, and Graph Neural Networks, to improve predictive capabilities (Zhang et al., 2017; Shi et al., 2015; Lan et al., 2022). However, these methods heavily depend on large-scale training data, such as extensive vehicle trip records or auxiliary weather information, which are often unavailable in real-world scenarios (De Montjoye et al., 2013; Zheng et al., 2008; Wang et al., 2018). Consequently, there has been an increasing focus on transfer learning to improve traffic flow prediction in data-limited cities (Yao et al., 2019a; Wang et al., 2021; Fang et al., 2022). These

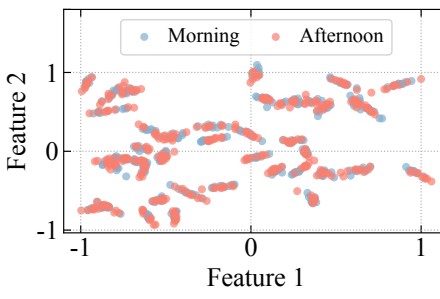 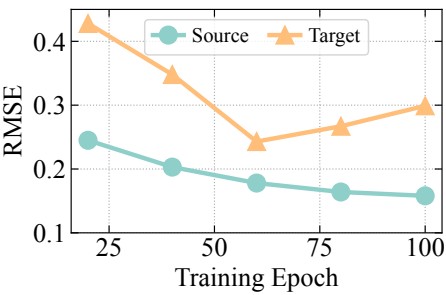

Figure 1: Limitations of existing methods. (left) Feature distribution of NYCBike data by STAN, with blue for morning peak (8:00-10:00) and red for afternoon (14:00-16:00) average flow. (right) Loss curves of source and target cities during the DCBike to DCTaxi transfer task using STAN.

methods generally involve training models using supervised learning or meta-learning on the source city, which has abundant data, and then fine-tuning the models on the target city with limited data.

However, existing methods still exhibit certain limitations: **i) Limited discrimination of extracted spatio-temporal features.** As shown in Figure 1 (left), we visualize the dimensionality-reduced features of morning and afternoon traffic patterns extracted from the NYCBike dataset using STAN (where "Feature 1" and "Feature 2" denote two dimensions of the combined features). The results reveal that different traffic patterns are poorly separated in the feature space, indicating that the learned spatio-temporal representations lack precision and discriminative power. **ii) Negative knowledge transfer.** In the DCBike to DCTaxi using STAN, the loss of the source city decreases steadily and converges during training, while the loss of the target city continues to increase (see Figure 1 (right)). This suggests that irrelevant or harmful information from the source city may be transferred to the target city, resulting in performance degradation due to negative transfer (Wang et al., 2019b).

To address these challenges, we novelly propose **A h**ierarchical **a**daptive **Trans**fer Learning Framework, i.e., AhaTrans. For the first challenge, we introduce a Spatio-Temporal Contrastive Embedding (STCE) module *at the feature level*. By enhancing the similarity of samples within the spatio-temporal feature space, STCE encourages the learned representations to better align with actual traffic flow, thus facilitating more accurate and discriminative spatio-temporal feature learning. According to the second challenge, we design a Guarded Transfer Experts Network (GTEN) *at the architectural level*, which explicitly distinguishes between shared and city-specific experts. This approach enables the target city to effectively leverage informative knowledge from the source city while mitigating the influence of potentially harmful information. *At the data adaptation level*, we propose a Transfer-based Reweighting (TBR) strategy for dynamic weighting of source city samples. In contrast to CrossTReS (Jin et al., 2022), which applies weighting solely along the spatial dimension, TBR jointly considers both temporal and spatial similarities. This enables robust cross-task transfer even between cities with similar spatial configurations. In summary, the main contributions are as follows:

- We design STCE to integrate contrastive learning into cross-city traffic flow prediction tasks, aiming to learn more precise and discriminative spatio-temporal representations. STCE achieves effective feature learning without relying on extensive data or detailed road network topologies, significantly reducing the costs of data acquisition and preprocessing. Furthermore, STCE establishes a generalized contrastive learning framework for spatio-temporal data, adaptively handling spatial and temporal variations across different cities.

- We alleviate the negative transfer issue from two complementary perspectives: model architecture and data adaptation. Specifically, during the transfer process, TBR accounts for temporal and spatial similarities between source and target cities to facilitate the acquisition of valuable knowledge. In addition, to prevent interference among different knowledge sources, GTEN employs a decoupled expert mechanism that allows the target city to effectively leverage useful knowledge from the source city while suppressing potentially harmful transfer.

- Extensive experimental results demonstrate that AhaTrans significantly outperforms existing methods, thereby validating its effectiveness in cross-city traffic flow prediction. We conduct comprehensive evaluations on multiple real-world datasets, including main experiments, ablation studies, hyperparameter analysis, efficiency comparisons, case studies, and generalization studies, systematically verifying our framework and the superiority of our overall approach.

## 2 PRELIMINARIES

### 2.1 NOTATIONS AND DEFINITIONS

**Definition 1 (Region).** Following previous research (Fang et al., 2022), we divide a city $c$ into a grid of size $h \times w$, e.g., $16 \times 16$, based on latitude and longitude. Each cell in this grid represents a specific region $r_{i,j}$, which is located at the $i$-th row and $j$-th column. Collectively, all regions in this grid form a complete region image of the city, denoted by $R_c = \{r_{1,1}, \cdots, r_{i,j}, \cdots, r_{h,w}\}$.

**Definition 2 (Inflow/Outflow).** In urban settings, each region $r_{i,j}$ experiences two types of flows, i.e., inflow and outflow, within a specified time interval $t$. These flows can be calculated using historical data of vehicle trajectories:

$$x_t^{(r_{i,j},in)} = \sum_{\tau \in \mathcal{T}} \left| g_{t-1} \notin r_{i,j} \text{ and } g_t \in r_{i,j} \right|, \qquad x_t^{(r_{i,j},out)} = \sum_{\tau \in \mathcal{T}} \left| g_{t-1} \in r_{i,j} \text{ and } g_t \notin r_{i,j} \right|. \quad (1)$$

Here, $\mathcal{T}$ is the set of all trajectories, where each trajectory $\tau = \{g_1, \cdots, g_t, \cdots, g_n\}$ spans $n$ GPS locations, with $g_t$ indicating the location at time $t$ for that trajectory.

**Definition 3 (Urban Traffic Flow Image).** For a given region set $R = \{r_{1,1}, \cdots, r_{i,j}, \cdots, r_{h,w}\}$, which represents a city, we define an "urban traffic flow image" for any time interval $t$ by combining inflow and outflow values across all regions. This urban flow image is represented as $X_t \in \mathbb{R}^{h \times w \times 2}$.

**Definition 4 (Urban Traffic Flow Image Time-Series).** Given the latest time interval $t$ and a historical time range $k$, we denote $\mathcal{X} = \{X_{t-k+1}, \cdots, X_{t-1}, X_t\} \in \mathbb{R}^{k \times h \times w \times 2}$ as the "urban traffic flow image time-series", where $\mathcal{X}(i, j, t, *)$ represents the inflow and outflow in region $r_{i,j}$ during $t$.

### 2.2 PROBLEM FORMULATION

When sufficient data is available in a *source* city but limited data in a *target* city (i.e., $|\mathcal{X}^{source}| \gg |\mathcal{X}^{target}|$), the objective of cross-city traffic flow prediction is to develop a function $f(\cdot, \cdot)$ that forecasts future patterns in all regions of the target city $T$ for the upcoming time interval $t + 1$:

$$\min_f \sum_{t+1} \text{Loss} \left( Y_{t+1}^{target}, \hat{Y}_{t+1}^{target} \right), \qquad s.t. \quad \hat{Y}_{t+1}^{target} = f \left( \mathcal{X}^{source}, \mathcal{X}^{target} \right), \quad (2)$$

where $Y_{t+1}^{target}$ and $\hat{Y}_{t+1}^{target}$ represent the actual observed data and predicted traffic flow values, respectively. The function $Loss(\cdot, \cdot)$, measuring prediction accuracy, can be calculated using metrics like Root Mean Squared Error (RMSE), Mean Absolute Error (MAE), and similar approaches.

## 3 METHODOLOGY

### 3.1 FRAMEWORK OVERVIEW

We propose AhaTrans, a novel hierarchical adaptive transfer learning framework for cross-city traffic flow prediction, as depicted in Figure 2. AhaTrans comprises three core components: a Guarded Transfer Experts Network, a Spatio-Temporal Contrastive Embedding module, and a Transfer-based Reweighting module, which jointly enhance the system in terms of model architecture, feature representation, and data adaptation. For better understanding, we provide detailed specifications of the model's input-output dimensions in Appendix B.2 and the pseudocode in Appendix B.3.

### 3.2 GUARDED TRANSFER EXPERTS NETWORK

As shown in Figure 2 (left), GTEN explicitly differentiates between a shared layer and city-specific layers, defining three types of experts—source, shared, and target—and two cities (source and target). The shared expert is responsible for learning common patterns, while the city-specific experts extract features unique to each city effectively. Specifically, the source expert is trained exclusively on source city data, the target expert models on the target city data, and the shared expert learns common patterns from both source and target city data. To further enhance the model's adaptability and overall performance, we introduce a gating network that selectively fuses the outputs of the various experts, depending on the specific characteristics of each city. This design allows the linear head network for each city to leverage the combined knowledge acquired from both the shared and city-specific experts for prediction. As illustrated in Figure 2 (left), the gating network employs a single-layer feedforward architecture and uses the SoftMax function as its activation. Specifically, the gating network takes the

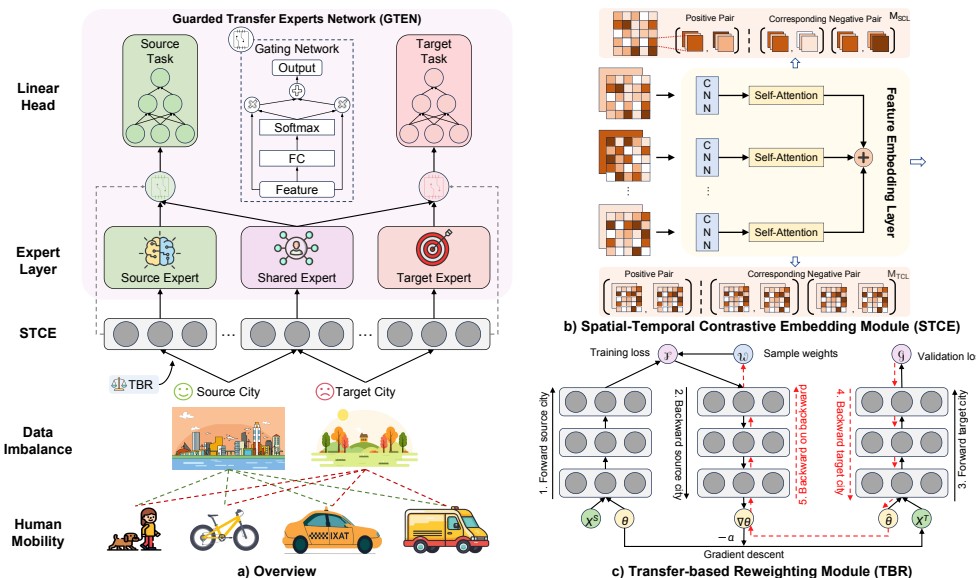

Figure 2: The framework of AhaTrans. a) Overview: This figure illustrates cross-city traffic flow prediction under data imbalance, emphasizing how GTEN differentiates between shared and city-specific experts. This design mitigates the influence of negative knowledge from the source city while facilitating the transfer of useful knowledge to the target city. b) STCE: Incorporates contrastive learning to enhance the discriminability of spatio-temporal features, thereby improving representation effectiveness. c) TBR: By dynamically reweighting source city samples, this component facilitates the extraction of knowledge most relevant to traffic patterns in the target city from the source.

feature representation $x$ as input and outputs weights for each expert. For city $c$, where $c$ can take the value of either $source$ or $target$, the fused output can be expressed as:

$$g^c(x) = w^c Expert^c(x) + w^{\text{shared}} Expert^{\text{shared}}(x), \qquad (3)$$

where $Expert^c(x)$ and $Expert^{\text{shared}}(x)$ are features from the city-specific and shared experts, respectively, weighted by $w^c$ and $w^{\text{shared}}$ as computed below:

$$w^c, w^{shared} = \text{Softmax}(FC(x)). \qquad (4)$$

Finally, the prediction result for city $c$ is expressed as follows:

$$y^c(x) = \text{LinearHead}^c(g^c(x)), \qquad (5)$$

where $\text{LinearHead}^c$ denotes the linear head corresponding to city $c$. Let $y^c_{\text{true}}$ denote the ground truth for city $c$; then, for each city $c$ the prediction loss is defined as

$$L_P^c = Loss\big(y^c(x), y^c_{\text{true}}\big). \qquad (6)$$

Here, $Loss(\cdot, \cdot)$ denotes MAE, which is used in our implementation as the loss function.

To theoretically support the knowledge isolation mechanism in GTEN, we present the following generalization bound(The proof and more theoretical support can be found in Appendix C.1):

**Theorem 3.1** (Knowledge Isolation Guarantee). *The expert separation structure of GTEN ensures the following upper bound on the generalization error for the target city:*

$$R_{\mathcal{D}_t}(h_t) \le \hat{R}_{\mathcal{D}_t}(h_t) + \Omega(m_t) + w^{shared} \cdot \eta_{s,t}, \qquad (7)$$

*where $\Omega(m_t)$ is the generalization gap related to the number of target city samples, $\eta_{s,t}$ is the transfer error from source city to target city, and $w^{shared}$ is the weight of the shared expert.*

This bound captures the trade-off between knowledge reuse and isolation: smaller transfer error $\eta_{s,t}$ and gating weight $w^{\text{shared}}$ reduce harmful transfer risk. GTEN ensures target cities benefit from shared knowledge while maintaining robustness against irrelevant or negative source patterns.

### 3.3 SPATIAL-TEMPORAL CONTRASTIVE EMBEDDING

#### 3.3.1 FEATURE EXTRACTION

To capture both spatial and local temporal patterns in traffic flow, we adopt a multi-head convolutional self-attention mechanism (Liu et al., 2020), following (Fang et al., 2022). Specifically, historical

data from $P$ previous days is used, where $L$ intervals before the current time step are selected from both source and target cities. The input is defined as $\mathcal{X} = \{X^1; X^2; \ldots; X^P\} \in \mathbb{R}^{L \times P \times h \times w \times 2}$,, where each $X^p = \{X_{(p,T-L)}, \ldots, X_{(p,T)}\}$. This tensor is fed into the multi-head convolutional self-attention layer. For each time step $X^i$, a convolutional subnetwork produces the query $Q^i$, key $K^i$, and value $V^i$ maps. Attention scores are computed via a compatibility function $M_\theta$:

$$S_{ij} = M_\theta(Q^i, K^j), \quad \alpha_{ij} = \frac{\exp(S_{ij})}{\sum_{j'=1}^{P} \exp(S_{ij'})}, \quad V^i = \sum_{j=1}^{P} \alpha_{ij} V^j.$$

For $H$ attention heads, the final output is

$$V^{\mathrm{MH},i} = \mathrm{Concat}(V^{(1),i}, V^{(2),i}, \ldots, V^{(H),i}). \tag{8}$$

This multi-head structure enhances the model's ability to extract diverse patterns in complex scenarios.

### 3.3.2 SPATIAL AND TEMPORAL CONTRASTIVE LEARNING

To boost both the expressiveness and discriminative strength of our spatio-temporal representations, we adopt a contrastive-learning framework (Hadsell et al., 2006) and train the network with the Rank-N-Contrast objective (Zha et al., 2024), which explicitly orders positives against a spectrum of hard negatives, yielding finer-grained feature separation.

**Spatial Contrastive Learning (SCL).** Given a batch of $B$ samples, we randomly select $N$ grids from each $h \times w$ spatial map, yielding spatial feature embeddings $\mathbb{V}^{sp} = \{\mathbf{V}_1^{sp}, \mathbf{V}_2^{sp}, \ldots, \mathbf{V}_B^{sp}\} \in \mathbb{R}^{L \times P \times 2}$, where $\mathbf{V}_i^{sp}$ denotes the spatial features of the $i$th sample. For any pair of samples, we treat $\mathbf{V}_m^{sp}$ as the anchor and $\mathbf{V}_n^{sp}$ as a comparison. We define $S_{n,m}^{sp}$ as the set of samples whose label distance is greater than or equal to that between $\mathbf{V}_m^{sp}$ and $\mathbf{V}_n^{sp}$. The similarity between $\mathbf{V}_m^{sp}$ and $\mathbf{V}_n^{sp}$, normalized via softmax over $S_{n,m}^{sp}$, is expressed as:

$$P(\mathbf{V}_n^{sp}|\mathbf{V}_m^{sp}, S_{n,m}^{sp}) = \exp\left(\mathrm{sim}(\mathbf{V}_m^{sp}, \mathbf{V}_n^{sp})/\tau\right) \Big/ \sum_{\mathbf{V}_k^{sp} \in S_{n,m}^{sp}} \exp\left(\mathrm{sim}(\mathbf{V}_m^{sp}, \mathbf{V}_k^{sp})/\tau\right). \tag{9}$$

Here, $\mathrm{sim}(\cdot, \cdot)$ is a similarity metric (e.g., negative L2 norm), and $\tau$ is a temperature parameter. For each anchor $\mathbf{V}_m^{sp}$, the loss is computed as the average negative log-likelihood over all other $BN - 1$ samples. The overall loss is then calculated by averaging across all anchors:

$$L_{SCL}^{(m)} = \sum_{n=1, n \neq m}^{BN} -\log\left(P(\mathbf{V}_n^{sp} \mid \mathbf{V}_m^{sp})\right)/(BN - 1), \quad L_{SCL} = \sum_{m=1}^{BN} L_{SCL}^{(m)}/BN. \tag{10}$$

**Temporal Contrastive Learning (TCL).** For each sample, we randomly select $M$ historical data points per day, resulting in $B \times M$ new samples that are aggregated into a batch $\mathbf{V}^{tp} = \{\mathbf{V}_1^{tp}, \mathbf{V}_2^{tp}, \ldots, \mathbf{V}_B^{tp}\} \in \mathbb{R}^{h \times w \times 2}$. Similar to the spatial setting, we apply the Rank-N-Contrast Loss, which ranks all temporal features based on their label-space order to capture underlying temporal dependencies. For the detailed formulation, see Appendix B.1.

### 3.3.3 OVERALL LOSS FUNCTION.

To fully integrate feature learning into the model training process, we have developed a composite loss function comprising prediction loss, spatial contrast loss, and temporal contrast loss. The composite loss is defined as follows:

$$L = L_P^c + \beta L_{SCL}^c + \gamma L_{TCL}^c, \quad c \in \{source, target\}, \tag{11}$$

where $\beta$ and $\gamma$ control the weight of spatial and temporal contrastive loss.

### 3.3.4 CONVERGENCE ANALYSIS

To ensure the stability and effectiveness of the learned features, we analyze the convergence behavior of the STCE module. The proof and more theoretical support can be found in Appendix C.2.

**Theorem 3.2** (Convergence Guarantee). *Given a sufficient number of training samples and an appropriate learning rate, spatio-temporal contrastive learning in STCE converges to a local optimum and guarantees that the learned feature representations have sufficient discriminative power.*

## 3.4 TRANSFER-BASED REWEIGHTING

At the data level, we adopt the Transfer-Based Reweighting (TBR) approach for the source city. This method assigns distinct weights to source city samples, enabling the model to prioritize information that aligns with the target city's data distribution during training. Unlike traditional approaches that

apply uniform weights across all samples, TBR shifts the model's training objective from minimizing a standard loss function to minimizing a weighted loss function:

$$\theta^*(W) = \arg\min_\theta \sum_{i=1}^{Q_s} W_i L_i^{source}(\theta), \tag{12}$$

where $Q_s$ denotes the total number of source city samples, $\theta$ represents the model parameters, and $W_i$ and $L_i^{source}(\theta)$ denote the weight and loss of the $i$-th sample, respectively. Initially, each $W_i$ is treated as a learnable parameter. Based on validation results from evaluating the source city model on target city data, the optimal weights $W$ are determined as:

$$W^* = \arg\min_{W, W \geq 0} \frac{1}{Q_T} \sum_{i=1}^{Q_T} L_i^{source}(\theta^*(W)), \tag{13}$$

where $Q_T$ represents the total number of target city samples used for training.

**Gradient-guided Weight Update.** To accelerate the training process and refine weight allocation, we propose a gradient-guided weight update mechanism. In each training batch, we load equal amounts of data from both source and target cities, reusing data for target cities with scarce samples. We employ vanilla Stochastic Gradient Descent (SGD). At each step $T$, the model parameters are updated as:

$$\theta_{T+1} = \theta_T - \alpha \nabla \left( \frac{1}{B} \sum_{i=1}^{B} L_i^{source}(\theta_T) \right), \tag{14}$$

where $B$ is the batch size, $\alpha$ is the learning rate, and $L_i^{source}(\theta_T)$ is the loss for the $i$-th sample at step $T$. We introduce a perturbation parameter $\epsilon_i$ to adjust the weights of each source city sample, expressed as $L_{i,\epsilon}^{source}(\theta) = \epsilon_i L_i^{source}(\theta)$. The updated model parameters after perturbation are:

$$\hat{\theta}_{T+1}(\epsilon) = \theta_T - \alpha \sum_{i=1}^{B} \nabla_\theta L_{i,\epsilon}^{source}(\theta_T). \tag{15}$$

The optimal perturbation $\epsilon_T^*$ is obtained by minimizing the loss over target city samples:

$$\epsilon_T^* = \arg\min_\epsilon \frac{1}{Q_T} \sum_{i=1}^{Q_T} L_i^{source}(\hat{\theta}_{T+1}(\epsilon)). \tag{16}$$

Subsequently, a gradient descent step is performed on $\epsilon_T$ using target city samples to refine the output while ensuring non-negative weights:

$$u_{i,T} = -\eta \frac{\partial}{\partial \epsilon_{i,T}} \frac{1}{B} \sum_{i=1}^{B} L_i^{source}(\hat{\theta}_{T+1}(\epsilon)) \Big|_{\epsilon_i, T=0'}, \tag{17}$$

where $\eta$ denotes the learning rate, and $u_{i,T}$ represents the gradient step size used to update the perturbation $\epsilon_{i,T}$. Finally, to ensure proper scaling of weights, we perform normalization:

$$W_{i,T} = \frac{\hat{W}_{i,T}}{\sum_j \hat{W}_{j,T} + \delta \left( \sum_j \hat{W}_{j,T} \right)}, \tag{18}$$

where $\delta$ is a small constant added to prevent division by zero.

**Theoretical Analysis.** To validate the effectiveness of TBR—particularly its ability to reduce the distribution shift between the source and target domains—we further analyze the generalization capability of its reweighting strategy. The proof and more support can be found in Appendix C.3.

**Theorem 3.3** (Reweighting Generalization Bound). *Let $\mathcal{H}$ be a hypothesis space of VC-dimension $d$, and $L$ be a bounded loss function such that $0 \leq L(f_\theta(x), y) \leq M$. For a model $f_\theta$ learned through the reweighting mechanism, with probability at least $1 - \delta$, the following generalization bound holds:*

$$\mathbb{E}_{(x,y) \sim P_T}[L(f_\theta(x), y)] \leq \mathbb{E}_{(x,y) \sim P_S}[W(x,y)L(f_\theta(x), y)] + d_{\mathcal{H} \Delta \mathcal{H}}(P_S^W, P_T) + \lambda + \epsilon, \tag{19}$$

*where $P_S^W$ is the weighted source distribution, $d_{\mathcal{H} \Delta \mathcal{H}}$ is the $\mathcal{H}$-divergence, $\lambda$ is the risk of the ideal joint hypothesis, and $\epsilon$ is a complexity term dependent on the sample sizes $Q_S$ and $Q_T$.*

## 4 EXPERIMENTS

### 4.1 EXPERIMENTAL SETUP

**Datasets.** We utilize six widely used open-source urban traffic datasets: NYCBike, CHIBike, DCBike, DCTaxi, BJTaxi, and Chengdu. These datasets span different time periods ranging from

Table 1: Performance comparison of selected methods, detailed results of the remaining methods can be found in Appendix E.5. The best and second-best results are marked in **bold** and underlined.

| Method | | AhaTrans | | TransGTR | | CrossTReS | | STAN | | ST-DAAN | | STSGCN | | TGCN | |
|---|---|---|---|---|---|---|---|---|---|---|---|---|---|---|---|
| Metric | | RMSE | MAE | RMSE | MAE | RMSE | MAE | RMSE | MAE | RMSE | MAE | RMSE | MAE | RMSE | MAE |
| NYCBike → CHIBike | 7 days | **0.0216** | **0.0059** | 0.0241 | 0.0074 | 0.0234 | 0.0069 | 0.0268 | 0.0087 | 0.0317 | 0.0092 | 0.0399 | 0.0103 | 0.0440 | 0.0099 |
| | 15 days | **0.0207** | **0.0051** | 0.0237 | 0.0066 | 0.0227 | 0.0061 | 0.0215 | 0.0061 | 0.0263 | 0.0079 | 0.0327 | 0.0096 | 0.0333 | 0.0089 |
| | 30 days | **0.0195** | **0.0047** | 0.0219 | 0.0058 | 0.0212 | 0.0056 | 0.0209 | 0.0052 | 0.0248 | 0.0073 | 0.0311 | 0.0081 | 0.0292 | 0.0080 |
| | Avg | **0.0206** | **0.0052** | 0.0232 | 0.0066 | 0.0224 | 0.0062 | 0.0231 | 0.0067 | 0.0276 | 0.0081 | 0.0346 | 0.0093 | 0.0355 | 0.0089 |
| DCBike → NYCBike | 7 days | **0.0379** | **0.0125** | 0.0414 | 0.0132 | 0.0408 | 0.0136 | 0.0439 | 0.0157 | 0.0489 | 0.0168 | 0.0512 | 0.0216 | 0.0500 | 0.0179 |
| | 15 days | **0.0373** | **0.0121** | 0.0384 | 0.0127 | 0.0397 | 0.0129 | 0.0406 | 0.0133 | 0.0424 | 0.0132 | 0.0444 | 0.0154 | 0.0484 | 0.0168 |
| | 30 days | **0.0369** | **0.0119** | 0.0378 | 0.0122 | 0.0383 | 0.0125 | 0.0388 | 0.0121 | 0.0408 | 0.0123 | 0.0410 | 0.0129 | 0.0411 | 0.0150 |
| | Avg | **0.0374** | **0.0122** | 0.0392 | 0.0127 | 0.0396 | 0.0130 | 0.0411 | 0.0137 | 0.0440 | 0.0141 | 0.0455 | 0.0166 | 0.0465 | 0.0166 |
| NYCBike → DCBike | 7 days | **0.0276** | **0.0075** | 0.0349 | 0.0106 | 0.0313 | 0.0091 | 0.0335 | 0.0099 | 0.0352 | 0.0108 | 0.0373 | 0.0112 | 0.0374 | 0.0110 |
| | 15 days | **0.0270** | **0.0067** | 0.0317 | 0.0093 | 0.0292 | 0.0078 | 0.0302 | 0.0082 | 0.0328 | 0.0091 | 0.0334 | 0.0091 | 0.0337 | 0.0090 |
| | 30 days | **0.0267** | **0.0062** | 0.0295 | 0.0080 | 0.0275 | 0.0069 | 0.0281 | 0.0076 | 0.0280 | 0.0078 | 0.0293 | 0.0079 | 0.0303 | 0.0078 |
| | Avg | **0.0271** | **0.0068** | 0.0320 | 0.0093 | 0.0293 | 0.0079 | 0.0306 | 0.0086 | 0.0320 | 0.0092 | 0.0333 | 0.0094 | 0.0338 | 0.0093 |
| DCBike → DCTaxi | 7 days | **0.0280** | **0.0056** | 0.0318 | 0.0068 | 0.0355 | 0.0073 | 0.0327 | 0.0085 | 0.0404 | 0.0106 | 0.0383 | 0.0109 | 0.0402 | 0.0109 |
| | 15 days | **0.0261** | **0.0052** | 0.0301 | 0.0065 | 0.0318 | 0.0069 | 0.0278 | 0.0061 | 0.0337 | 0.0078 | 0.0334 | 0.0081 | 0.0331 | 0.0083 |
| | 30 days | **0.0254** | **0.0049** | 0.0277 | 0.0053 | 0.0289 | 0.0057 | 0.0264 | 0.0054 | 0.0309 | 0.0067 | 0.0307 | 0.0066 | 0.0312 | 0.0069 |
| | Avg | **0.0265** | **0.0052** | 0.0299 | 0.0062 | 0.0321 | 0.0066 | 0.0290 | 0.0067 | 0.0350 | 0.0084 | 0.0341 | 0.0085 | 0.0348 | 0.0087 |

several months to one year and have been extensively used in related research. Table 4 presents the statistical information of these datasets. For detailed information on the datasets, please refer to Appendix E.1. To assess the generalizability of the model for spatio-temporal knowledge transfer, we consider both intra-city tasks and inter-city scenarios. Specifically, at the $16 \times 16$ grid resolution, we designate NYCBike and DCBike as source cities, while ChicagoBike, NYCBike, DCBike, and DCTaxi serve as target cities. For experiments at the $32 \times 32$ grid resolution, we utilize BJTaxi as the source city and Chengdu as the target city to evaluate the model's performance across different spatial scales. The data preprocessing and split method can be found in Appendix D.

**Baselines.** We compare AhaTrans against fourteen state-of-the-art (SOTA) methods spanning four main categories: statistical learning, deep learning, transfer learning, and foundation models. (i) For statistical learning methods, we employ the ARIMA (Zhang, 2003) model as a baseline for modeling and predicting non-stationary time series. (ii) Regarding deep learning methods, we selected ConvLSTM (Shi et al., 2015), STResNet (Zhang et al., 2017), STSGCN (Song et al., 2020), and TGCN (Zhao et al., 2019) as our baselines. These models are initially pre-trained on source city data and subsequently fine-tuned on target city data. (iii) For the comparison of transfer learning methods, we selected RegionTrans (Wang et al., 2019a), MetaST (Yao et al., 2019a), ST-DAAN (Wang et al., 2021), STAN (Fang et al., 2022), CrossTReS (Jin et al., 2022), and TransGTR (Jin et al., 2023) as baselines for evaluation. (iv) For foundation models, we incorporated three representative methods: PatchTST (Nie et al., 2022), UrbanGPT (Li et al., 2024), and UniST (Yuan et al., 2024), to evaluate their capabilities in spatio-temporal prediction tasks. More detailed information in Appendix E.2.

**Implementation Details.** First, we segment the cities in the study area into predefined regions (or grids) and divide the temporal dimension into distinct, non-overlapping intervals (see Table 4). Next, we select three days of historical data, with each day comprising nine time intervals. Based on Equation 1, we compute the inflow and outflow for each region and normalize the traffic flow data to the range $[0, 1]$. For model training, the batch size, dropout rate, and learning rate are set to 32, 0.5, and $1 \times 10^{-6}$, respectively. All experiments are implemented using the PyTorch framework and executed on NVIDIA A100 80GB GPUs. For more detailed information, please refer to Appendix E.3 and visit our anonymous repository (`https://anonymous.4open.science/r/AhaTrans-A37F`).

## 4.2 PERFORMANCE COMPARISON

This section presents a comprehensive performance evaluation of AhaTrans against statistical learning, deep learning, and transfer learning baselines across multiple transfer tasks. Note that foundation model-based approaches follow a fundamentally different paradigm from the aforementioned three categories and are therefore discussed separately in Appendix E.5.2.

**Overall Performance.** We evaluate AhaTrans across five transfer tasks, with results averaged over three independent runs as presented in Tables 1, 2, and 5. The experimental results demonstrate that AhaTrans achieves significant performance improvements across all test scenarios. Across four standard transfer tasks employing $16 \times 16$ grid resolution, AhaTrans exhibits consistent advantages. Compared to the best-performing baseline CrossTReS (Jin et al., 2022), AhaTrans achieves average

Table 2: Performance comparison on BJTaxi → Chengdu task.

| Metric | MetaST | ST-DAAN | STAN |
|---|---|---|---|
| RMSE | 0.0390 | 0.0909 | 0.0345 |
| MAE | 0.0275 | 0.0557 | 0.0196 |

| Metric | CrossTReS | TransGTR | AhaTrans |
|---|---|---|---|
| RMSE | 0.0323 | 0.0341 | **0.0307** |
| MAE | 0.0187 | 0.0204 | **0.0168** |

Table 3: Comparison with different variants of AhaTrans. The best results are marked in **bold**.

| Method | AhaTrans | | w/o STCE | | w/o GTEN | | w/o TBR | |
|---|---|---|---|---|---|---|---|---|
| Metric | RMSE | MAE | RMSE | MAE | RMSE | MAE | RMSE | MAE |
| NYCBike → CHIBike | **0.0216** | **0.0059** | 0.0225 | 0.0062 | 0.0235 | 0.0067 | 0.0228 | 0.0063 |
| DCBike → NYCBike | **0.0379** | **0.0125** | 0.0388 | 0.0128 | 0.0405 | 0.0133 | 0.0392 | 0.0131 |
| NYCBike → DCBike | **0.0276** | **0.0075** | 0.0309 | 0.0085 | 0.0315 | 0.0089 | 0.0303 | 0.0081 |
| DCBike → DCTaxi | **0.0280** | **0.0056** | 0.0313 | 0.0087 | 0.0298 | 0.0064 | 0.0306 | 0.0072 |

reductions of **9.61%** in RMSE and **12.83%** in MAE. Against all transfer learning methods, improvements are more pronounced with RMSE and MAE reductions averaging **19.78%** and **25.84%**, respectively. Most remarkably, compared to traditional deep learning models without transfer learning, AhaTrans achieves breakthrough improvements of **54.01%** in RMSE and **80.88%** in MAE, demonstrating the critical importance of cross-city knowledge transfer.

To evaluate scalability across different grid resolutions, we evaluated the Beijing-to-Chengdu task using $32 \times 32$ high-resolution grids. Despite facing complex spatial structures and limited data, AhaTrans maintains its performance advantage, achieving improvements of **4.95%** and **10.16%** in RMSE and MAE, respectively, compared to CrossTReS (Table 2). These results validate the adaptability of AhaTrans across varying grid resolutions. More details are in Appendix F.2.1.

**Robustness in Data-Scarce Scenarios.** AhaTrans demonstrates superior robustness in data-scarce scenarios. In the NYCBike-to-CHIBike transfer task, when target city data is reduced from 30 days to 7 days, AhaTrans exhibits only a **10.77%** increase in RMSE compared to **17.99%** for other transfer learning methods and up to **74.51%** for non-transfer methods. These results indicate that AhaTrans effectively captures shared characteristics between cities and facilitates robust knowledge transfer under limited data conditions, making it particularly valuable for practical deployment scenarios.

## 4.3 MODEL ANALYSIS

### 4.3.1 ABLATION STUDY

To systematically assess the contributions of key components—STCE, GTEN, and TBR—to the performance of AhaTrans, we conducted a series of ablation studies by selectively removing each module. As shown in Table 3, the complete AhaTrans model consistently outperforms all its ablated variants across all datasets, underscoring the critical role each component plays within the overall framework. Furthermore, we carried out a more fine-grained ablation analysis focusing on the two core modules, STCE and GTEN, to examine the impact of spatio-temporal contrastive learning along both temporal and spatial dimensions, as well as to clarify the individual contributions of different expert networks. Detailed results of this analysis can be found in Appendix F.1.

### 4.3.2 HYPERPARAMETER SENSITIVITY

**Effect of Data Amount.** Under the default configuration, the source and target cities are assigned 12 months and 1 month of data, respectively. To examine the impact of data volume on model performance, we conduct a controlled experiment by varying the data quantity in the source city (left) and the target city (right), respectively. As shown in Figure 3, model performance exhibits a marked improvement with increasing data volume, aligning with expected trends. Notably, AhaTrans consistently achieves superior accuracy and demonstrates remarkable robustness across all settings.

**Tuning $\beta$ and $\gamma$.** We conduct a sensitivity analysis by adjusting the weights $\beta$ and $\gamma$ in Equation 11. We fix $\beta$ (or $\gamma$) and vary $\gamma$ (or $\beta$) from 0.1 to {0.3, 0.5, 0.7}. In Figure 4, the optimal setting for both $\beta$ and $\gamma$ is 0.1, and AhaTrans consistently outperforms the baselines under various weight preferences.

**Sensitivity to $P$ and $L$.** As shown in Figure 5, AhaTrans demonstrates stable performance under various settings of $P$ and $L$, further confirming its superior robustness in dynamic environments. In contrast, STAN shows greater sensitivity to parameter changes, which further highlights the advantages of AhaTrans in cross-city traffic flow prediction.

**Effect of MLP Layer Number.** Finally, we examine the impact of varying the number of MLP layers in the city-specific expert and linear head on performance, with configurations of 1, 2, 3, and 4 layers. As shown in Figure 6, the optimal number of MLP layers is 2. Too many layers may lead to overfitting, while too few hinder the model's ability to effectively capture complex patterns.

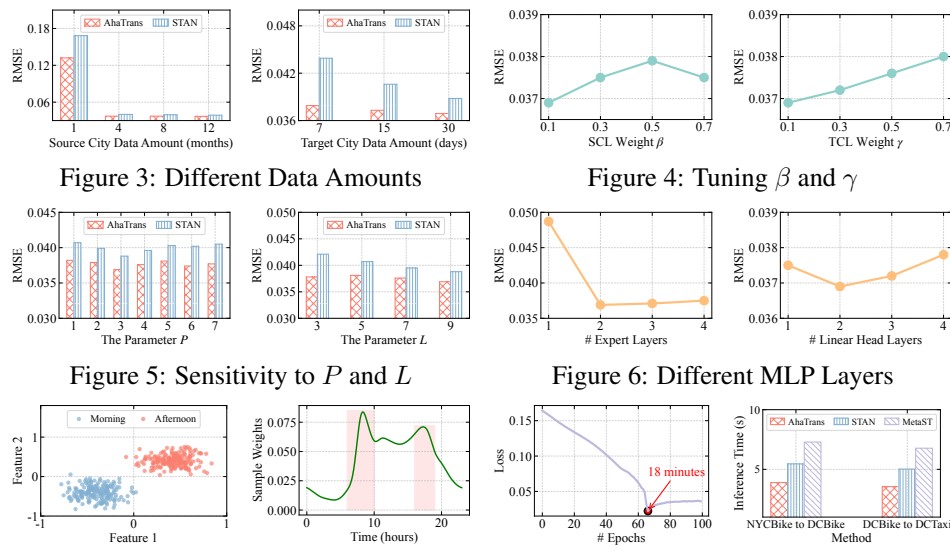

Figure 3: Different Data Amounts

Figure 4: Tuning $\beta$ and $\gamma$

Figure 5: Sensitivity to $P$ and $L$

Figure 6: Different MLP Layers

Figure 7: Case Study

Figure 8: Efficiency Evaluation

### 4.3.3 CASE STUDY

In the NYCBike-to-CHIBike transfer task, we employ the STCE to extract spatio-temporal features from the NYC Bike dataset, followed by dimensionality reduction, as illustrated in Figure 7 (left). Compared to Figure 1 (left), the learned features exhibit higher precision and stronger discrimination. As shown in Figure 7 (right), the learned weight distribution concentrates during peak traffic periods, reflecting similar traffic patterns between cities. This supports the effectiveness of the TBR module in capturing cross-city transferable knowledge. The spatial perspective case study is in Appendix F.4.

### 4.3.4 EFFICIENCY STUDY

Moreover, we assess model efficiency on the NYCBike-to-DCBike task. The training loss curve for AhaTrans in Figure 8 (left) demonstrates rapid convergence, with significant loss reduction within 60 epochs before stabilization. As shown in Figure 8 (right), AhaTrans achieves superior inference speed compared to existing methods, as it avoids the complex discrimination and multi-city learning processes required by other models. Comprehensive parameter analysis is presented in Appendix F.5.

### 4.3.5 GENERALIZATION STUDY

To validate the generalizability of the AhaTrans framework, we conducted additional experiments on cross-city crime prediction. Using the Chicago crime dataset as the source domain and the NYC crime dataset as the target domain, with only 10% training data available in the target domain to simulate data scarcity, AhaTrans achieves substantial reductions of 53.1% in RMSE and 39.2% in MAE compared to the best baseline method, demonstrating the broad applicability of our framework beyond traffic prediction (detailed results are in Appendix F.2.2).

## 5 CONCLUSION

In this paper, we present AhaTrans, a novel hierarchical adaptive transfer learning Framework designed to enhance discriminative spatio-temporal feature learning, optimize knowledge transfer effectiveness, and mitigate harmful transfer in cross-city traffic flow prediction. Specifically, the STCE module significantly enhances the discriminative capability of spatio-temporal features through contrastive learning, while the GTEN and TBR modules work synergistically to ensure efficient knowledge transfer from source to target cities while effectively suppressing harmful knowledge interference. Extensive experimental validation demonstrates that AhaTrans achieves substantial performance improvements over baselines. Moreover, AhaTrans exhibits superior computational efficiency by eliminating the need for complex data preprocessing steps or extensive computational resources. Comprehensive model analysis further reveals the remarkable robustness and superior generalization capability of AhaTrans across different scenarios, while detailed visualization experiments confirm the framework's strong interpretability, thereby providing a reliable and efficient solution for cross-city traffic flow prediction and other spatio-temporal prediction tasks.

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

# APPENDIX

# Contents

# A  RELATED WORKS

## A.1  URBAN TRAFFIC FLOW PREDICTION

Urban traffic flow prediction analyzes historical traffic data to forecast future spatio-temporal patterns, which is crucial for smart city development and traffic system optimization (Xie et al., 2020). Traditionally, statistical time series models, such as ARIMA (Zhang, 2003) and regression models incorporating spatio-temporal regularization, have been extensively used in this domain. However, these statistical approaches have limited capacity for learning and fail to adequately capture the complex spatio-temporal dependencies inherent in urban flow data. To improve prediction accuracy, deep learning methods, including convolutional neural networks (CNNs)(Zhang et al., 2017; 2019), recurrent neural networks (RNNs)(Shi et al., 2015; Zhou et al., 2018; Yao et al., 2019b), and graph neural networks (GNNs)(Li et al., 2017; Lan et al., 2022), have been widely employed (Liang et al., 2019; Pan et al., 2019; Wang et al., 2020). ConvLSTM (Shi et al., 2015) formulates precipitation forecasting as a spatio-temporal sequence prediction problem and utilizes convolutional LSTMs to model inherent dependencies. ST-ResNet (Zhang et al., 2017) adopts a residual neural network architecture with three specialized branches dedicated to processing distinct temporal properties—short-term proximity, periodicity, and long-term traffic trends—thereby improving prediction accuracy. Zhang et al.(Zhang et al., 2019) introduce a multi-task deep learning framework that jointly predicts node and edge flow in spatio-temporal networks, leveraging inter-task relationships to improve overall predictive performance. Despite their effectiveness, deep learning models rely heavily on extensive historical traffic data, which is often scarce in real-world urban environments. Moreover, multi-task learning approaches are prone to the "seesaw phenomenon," wherein improving the accuracy of one task may inadvertently degrade another's performance (Tang et al., 2020).

On the other hand, inspired by the success of pre-trained models in natural language processing, researchers have begun exploring the possibility of building universal spatio-temporal pre-trained models. Methods such as UniST (Yuan et al., 2024) and UrbanGPT (Li et al., 2024) leverage diverse spatio-temporal data from various scenarios for pre-training, enabling them to capture complex spatio-temporal dynamics and enhance generalization capabilities through knowledge-guided prompting mechanisms. The core advantage of these approaches lies in their ability to achieve effective predictions in few-shot or even zero-shot scenarios, providing new insights for addressing data scarcity issues in real-world urban sensing applications. However, these pre-training approaches still face numerous challenges: first, constructing and training large-scale pre-trained models requires substantial computational resources and costs; second, acquiring sufficiently large-scale and high-quality multi-source spatio-temporal data is itself a formidable task; moreover, the spatial knowledge alignment problem across different cities and scenarios remains a critical bottleneck constraining model generalization capabilities.

To address the challenge of traffic data scarcity in urban environments, researchers have increasingly leveraged transfer learning methods to facilitate cross-city knowledge transfer, thereby alleviating data insufficiency issues in target cities. We propose AhaTrans, a novel transfer learning framework specifically designed for this domain. AhaTrans adopts a dual-task architecture, but unlike traditional multi-task learning frameworks, it primarily focuses on enhancing prediction performance in target cities under data-scarce scenarios. Notably, the AhaTrans architecture demonstrates significant efficiency advantages, requiring only information from a single source city to achieve effective knowledge transfer. Compared to large-scale pre-trained models, it eliminates the need for massive computational resources and costs, providing a more economically feasible solution for practical applications.

## A.2  TRANSFER LEARNING FOR TRAFFIC PREDICTION

Data scarcity continues to be a prevalent challenge in traffic flow prediction, arising from disparities in urban modernization and constraints imposed by data privacy regulations. To address this issue, researchers have explored a range of deep learning-driven transfer learning strategies, such as RegionTrans (Wang et al., 2019a), MetaST (Yao et al., 2019a), ST-DAAN (Wang et al., 2021), STAN (Fang et al., 2022), CrossTReS (Jin et al., 2022), MGAT (Mo & Gong, 2022), and MetaCitta (Sao et al., 2023). RegionTrans (Wang et al., 2019a) enhances transferability by imposing similarity constraints on auxiliary data, while MetaST (Yao et al., 2019a) focuses on extracting and transferring long-term temporal patterns. ST-DAAN (Wang et al., 2021) leverages deep adaptive networks (DAN) for domain adaptation, enabling cross-city fine-tuning. However, most existing transfer learning approaches emphasize static knowledge transfer without accounting for the dynamic correlations

between cities. STAN (Fang et al., 2022) introduces an adaptive mechanism that dynamically adjusts to temporal variations in urban data, achieving superior performance in cross-city traffic prediction. MetaCitta (Sao et al., 2023) employs an adaptive learning strategy to integrate multiple data sources effectively, thereby improving prediction accuracy in novel environments and data-scarce scenarios. However, these studies primarily relied on Euclidean relationships between regions, overlooking the incorporation of complex semantic information. To address this limitation, Lu et al. (2022) proposed a model based on a graph data structure, where meta-knowledge was extracted using GRU and GAT modules. To effectively capture structural information, a graph construction loss was introduced. The node-level meta-knowledge was then utilized in a parameter generation module to produce non-shared parameters for the feature extractor. To further enhance adaptability across cities, TransGTR (Jin et al., 2023) was developed to enable the transfer of graph structures between different urban environments. Additionally, to mitigate the risk of negative transfer, Jin et al. (2022) introduced a selective cross-city transfer learning approach, which filters out detrimental source knowledge. Their method incorporated both edge-level and node-level adaptations to train the feature extraction network and employed a weighting network for loss computation. This method integrates edge-level and node-level adaptation mechanisms to train the feature extraction network and introduces a weighting network for loss computation, effectively capturing spatial similarities across cities. However, it overlooks the temporal dimension, which diminishes its performance advantage in cross-task transfer scenarios within cities that share similar spatial structures. In the context of multi-granular transfer learning, MGAT (Mo & Gong, 2022) employed multiple convolutional kernels to extract information at various granularities and leveraged an attention mechanism to facilitate knowledge sharing across cities. Nevertheless, their approach remained limited to modeling Euclidean relationships among regions, without incorporating more complex semantic dependencies.

Leveraging the abundant spatio-temporal information embedded within samples, AhaTrans implements a sample reweighting mechanism that provides a more precise characterization of the knowledge transfer process, thus ensuring consistent performance across both cross-city and intra-city prediction tasks. Furthermore, unlike existing approaches, AhaTrans achieves enhanced computational efficiency without relying on computationally intensive multi-scale convolution operations, while maintaining comparable predictive accuracy. Notably, the sample reweighting strategy represents just one of several innovative components within the AhaTrans framework. The architecture also incorporates the GTEN module, which selectively assimilates valuable knowledge patterns from source cities while filtering out potentially detrimental information, thereby significantly improving the robustness of the cross-city knowledge transfer process.

### A.3 CONTRASTIVE LEARNING FOR FEATURE EXTRACTION

Contrastive learning enhances representation learning by capturing both similarities and differences between samples, thereby substantially improving performance in diverse machine learning tasks (Wu et al., 2018; Zeng et al., 2021). However, existing urban flow prediction methods predominantly adopt an end-to-end training paradigm without explicitly optimizing spatio-temporal feature learning. InstDisc (Wu et al., 2018) enhances model discriminability by reducing intra-instance representation distances while maximizing inter-instance separation. Zeng et al. (2021) employs intra-domain intent clustering to minimize intra-class variance and maximize inter-class variance, thereby capturing fine-grained semantic features more effectively.

Recent research has explored the integration of contrastive learning into traffic flow prediction tasks. ST-SSL (Ji et al., 2023) implements a self-supervised contrastive learning approach that enhances spatio-temporal graph representations. Through clustering techniques, this method effectively preserves the spatial heterogeneity across different functional urban areas, primarily benefiting traffic prediction in data-rich environments. Similarly, STCL-AGA (Zhang et al., 2024) introduced a spatio-temporal contrastive learning framework featuring dynamic graph structures. By incorporating a flow-aware node masking mechanism, it successfully captures evolving traffic patterns, thus enhancing predictive performance. Other approaches such as STGL (Zhan et al., 2025) and UrbanGCL (Pan et al., 2023) rely on detailed road network topologies, conducting contrastive learning at node or graph levels, but require manually constructed graphs that must be rebuilt when city infrastructure changes—a significant limitation for dynamic urban environments.

Nevertheless, these methodologies face several critical limitations. First, existing methods predominantly rely on graph-structured data requiring detailed road network topologies, making them less adaptable to cities with different infrastructure layouts or when topology changes over time. Second, most current approaches target single-city traffic prediction in data-abundant scenarios, failing to

address the fundamental challenge of cross-city transfer learning under data scarcity conditions. Third, prior works do not adequately address the significant spatio-temporal heterogeneity between cities, a crucial factor in cross-city prediction tasks.

To address these limitations, our AhaTrans introduces the STCE module, which develops a contrastive learning mechanism specifically designed for generalized prediction under data-constrained conditions. This innovative module offers several key contributions. Unlike graph-dependent approaches, STCE is specifically designed for grid-based spatio-temporal data, requiring no graph construction. This makes it more versatile and efficient, particularly for integrating with raster-format data such as weather and remote sensing inputs. Furthermore, STCE is tailored for cross-city transfer learning under data scarcity, specifically addressing the spatio-temporal heterogeneity between different urban environments.

## B  ADDITIONAL DETAILS ON METHODS

### B.1  TEMPORAL CONTRASTIVE LEARNING IN STCE

For each sample, we randomly select $M$ data points from its historical data on a daily basis. Consequently, $B \times M$ new samples are generated each day and aggregated into a new batch, denoted as $\mathbf{V}^{tp} = \{\mathbf{V}_1^{tp}, \mathbf{V}_2^{tp}, \dots, \mathbf{V}_B^{tp}\} \in \mathbb{R}^{h \times w \times 2}$. In a manner analogous to our treatment of spatial dimensions, we employ the Rank-N-Contrast Loss. This method orders the complete set of temporal features according to their sequence in the label space, thereby capturing the inherent temporal relationships among samples. The corresponding equations can be represented as following:

$$P(\mathbf{V}_n^{tp}|\mathbf{V}_m^{tp}, S_{n,m}^{tp}) = \frac{\exp\left(\frac{\text{sim}(\mathbf{V}_m^{tp}, \mathbf{V}_n^{tp})}{\tau}\right)}{\sum_{\mathbf{V}_k^{tp} \in S_{n,m}^{tp}} \exp\left(\frac{\text{sim}(\mathbf{V}_m^{tp}, \mathbf{V}_k^{tp})}{\tau}\right)} \tag{20}$$

$$L_{TCL}^{(m)} = \frac{1}{BM - 1} \sum_{n=1, n \neq m}^{BM} -\log\left(P(\mathbf{V}_n^{tp}|\mathbf{V}_m^{tp}, S_{n,m}^{tp})\right) \tag{21}$$

$$L_{TCL} = \frac{1}{BM} \sum_{m=1}^{BM} L_{TCL}^{(m)} \tag{22}$$

### B.2  MODULE ARCHITECTURE AND TENSOR DIMENSIONS

The AhaTrans framework processes input tensors with dimensions $(B, P, L, H, W, C) = (32, 3, 9, 16, 16, 2)$, where $B$ represents the batch size, $P$ denotes the number of historical days, $L$ indicates the time intervals per day, $H \times W$ corresponds to the spatial grid dimensions, and $C$ represents the number of channels.

In the Spatio-Temporal Context Encoder (STCE) module, we first merge the day $(P)$ and time $(L)$ dimensions into a unified temporal sequence of length $T = PL = 27$, yielding a tensor of shape $(32, 27, 16, 16, 2)$. Subsequently, we apply a $3 \times 3$ convolutional operation to generate the Query $(Q)$, Key $(K)$, and Value $(V)$ feature maps, transforming the tensor to shape $(32, 27, 16, 16, 1024)$. The multi-head attention mechanism operates along the temporal dimension $T$, maintaining consistent input and output shapes of $(32, 27, 1024)$. Following sequence aggregation, the STCE module outputs a feature vector $\mathbf{x}$ with dimensions $(32, 1024)$.

Within the Gated Transfer Expert Network (GTEN) module, the expert networks transform the feature vector $\mathbf{x} \in \mathbb{R}^{32 \times 1024}$ to $\mathbb{R}^{32 \times 512}$, while the gating network produces a weight vector of shape $(32, 2)$, corresponding to the city-specific and shared experts. The prediction head processes the fused feature representation $(32, 512)$ through a two-layer Multi-Layer Perceptron (MLP) with architecture $512 \rightarrow 256 \rightarrow 512$, resulting in a tensor of shape $(32, 512)$. In the final stage, this output is reshaped into a spatial grid of dimensions $(32, 16, 16, 2)$ to produce the model's prediction.

### B.3  PSEUDOCODE OF AHATRANS

The detailed pseudocode of AhaTrans is shown in Algorithm 1.

---

**Algorithm 1:** Traffic Flow Prediction with AhaTrans

---

**Input:** source and target traffic flow data $\mathcal{D}^{source}, \mathcal{D}^{target}$, Batch size $B$, Learning rate $\alpha$,
    Spatio-temporal weights $\beta, \gamma$
**Output:** Model parameters $\theta$

1  Initialize model parameters $\theta$;
2  **for** $\mathcal{B}^{source} \in \mathcal{D}^{source}$ and $\mathcal{B}^{target} \in \mathcal{D}^{target}$ **do**
    // *** Source/Target Forward Processing ***
3    **for** $c \in \{source, target\}$ **do**
4     $\mathcal{B}^c \leftarrow$ FeatureEmbedding($\mathcal{B}^c$);
5     $L^c_{SCL}, L^c_{TCL} \leftarrow$ SCL($\mathcal{B}^c$), TCL($\mathcal{B}^c$);
6     **for** $(x, y) \in \mathcal{B}^c$ **do**
7      $E^c, E^{shared} \leftarrow$ Expert$^c(x)$, Expert$^{shared}(x)$;
8      $w^c, w^{shared} \leftarrow$ Softmax(Gate($x$));
9      $g^c \leftarrow w^c \cdot E^c + w^{shared} \cdot E^{shared}$;
10      $\hat{y}^c \leftarrow$ LinearHead$^c(g^c)$;
11      Compute sample loss: $L^c_i \leftarrow Loss(y, \hat{y}^c)$;
12      **if** $c = source$ **then**
13       $\epsilon_i \leftarrow 0, L^c_i \leftarrow \epsilon_i L^c_i$;
14      **end**
15     **end**
16     $L^c_P \leftarrow \frac{1}{B} \sum_{i \in \mathcal{B}^c} L^c_i$;
17     $L^c \leftarrow L^c_P + L^c_{SCL} + L^c_{TCL}$;
18     $\nabla \theta_c \leftarrow$ Backward($L^c, \theta_c$);
19     $\hat{\theta}_c \leftarrow \theta_c - \alpha \cdot \nabla \theta_c$;
20    **end**
    // *** Target Validation Processing ***
21    **for** $(x, y) \in \mathcal{B}^{target}$ **do**
22     $x \leftarrow$ FeatureEmbedding($x, \hat{\theta}$);
23     $E^{source}_{val}, E^{shared}_{val} \leftarrow$ Expert$^{source}(x, \hat{\theta})$, Expert$^{shared}(x, \hat{\theta})$;
24     $w^{source}_{val}, w^{shared}_{val} \leftarrow$ Softmax(Gate($x, \hat{\theta}$));
25     $g_{val} \leftarrow w^{source}_{val} \cdot E^{source}_{val} + w^{shared}_{val} \cdot E^{shared}_{val}$;
26     $\hat{y}_{val} \leftarrow$ LinearHead$^{source}(g_{val})$;
27     Compute sample loss: $L_{(i,val)} \leftarrow Loss(y, \hat{y}_{val})$;
28     $L_{(P,val)} \leftarrow \frac{1}{B} \sum_{i \in \mathcal{B}^{target}} L_{(i,val)}$;
29     $L_{val} \leftarrow L_{(P,val)} + L^{target}_{SCL} + L^{target}_{TCL}$;
30    **end**
31    $\nabla \epsilon \leftarrow$ Backward($L_{val}, \epsilon$);
32    $\hat{W} \leftarrow \max(-\nabla \epsilon, 0), W = \frac{\hat{W}}{\sum_{k \in \mathcal{B}^{source}} \hat{W} + \delta \sum_{k \in \mathcal{B}^{source}} \hat{W}}$;
33    $\hat{L}^{source}_P \leftarrow \sum_{i \in \mathcal{B}^{source}} W_i \cdot L^{source}_i$;
34    $\hat{L}^{source} \leftarrow \hat{L}^{source}_P + L^{source}_{SCL} + L^{source}_{TCL}$;
    // *** Parameter Update ***
35    $\nabla \theta^{source} \leftarrow$ Backward($\hat{L}^{source}, \theta$);
36    $\nabla \theta^{target} \leftarrow$ Backward($L^{target}, \theta$);
37    $\theta \leftarrow$ OptimizorStep($\theta, \nabla \theta^{source \cup target}$);
38  **end**

---

## C   THEORETICAL FOUNDATIONS OF MODULE DESIGN

### C.1   THEORETICAL ANALYSIS OF GTEN

In this section, we present a theoretical analysis of the Guarded Transfer Experts Network (GTEN) to demonstrate its effectiveness in facilitating beneficial knowledge transfer while mitigating harmful interference in cross-city traffic flow prediction.

### C.1.1 PROBLEM FORMULATION AND ASSUMPTIONS

Let us denote the source city data distribution as $\mathcal{D}_s$ and the target city data distribution as $\mathcal{D}_t$, where $(x, y) \sim \mathcal{D}_c$ represents input features and traffic flow labels from city $c \in \{s, t\}$. We make the following assumptions:

**Assumption C.1** (Distribution Shift). *$\mathcal{D}_s \neq \mathcal{D}_t$, but there exists shared structure that makes knowledge transfer possible.*

**Assumption C.2** (Shared Feature Representation). *There exists a feature space $\mathcal{Z}$ and an encoder $\phi : \mathcal{X} \to \mathcal{Z}$ such that partial knowledge can be transferred between cities.*

**Assumption C.3** (Expert Knowledge Decomposition). *The traffic flow prediction task can be decomposed into city-specific components and shared components.*

**Assumption C.4** (Sufficient Data). *Source city data is abundant, while target city data is limited but sufficient to learn city-specific patterns.*

The GTEN model consists of three types of experts: source city expert $E_s$, target city expert $E_t$, and shared expert $E_{shared}$. For an input $x$, each expert produces a hidden representation in $\mathbb{R}^d$. The gating network computes weights for each expert, and the final prediction is generated through a linear head network.

### C.1.2 GENERALIZATION ERROR BOUNDS WITH EXPERT SEPARATION

**Lemma C.5** (Expert Separation Generalization Bound). *Under the GTEN framework, the expected risk for the target city $R_{\mathcal{D}_t}(h_t)$ is bounded by:*

$$R_{\mathcal{D}_t}(h_t) \leq \hat{R}_{\mathcal{D}_t}(h_t) + \sqrt{\frac{\log(1/\delta)}{2m_t}} + \lambda \cdot d_{\mathcal{H}\Delta\mathcal{H}}(\mathcal{D}_s, \mathcal{D}_t) \tag{23}$$

*where $\hat{R}_{\mathcal{D}_t}$ is the empirical risk on the target domain, $m_t$ is the number of target domain samples, $\delta$ is a confidence parameter, $\lambda$ reflects the contribution of source domain knowledge to the target domain, and $d_{\mathcal{H}\Delta\mathcal{H}}$ is the $\mathcal{H}$-divergence between the two distributions.*

*Proof.* We first decompose the target city prediction function as:
$$h_t(x) = w^t \cdot E^t(x) + w^{shared} \cdot E^{shared}(x) \tag{24}$$
The true risk for the target city can be expressed as:
$$R_{\mathcal{D}_t}(h_t) = \mathbb{E}_{x \sim \mathcal{D}_t}[\ell(h_t(x), f_t(x))] \tag{25}$$
Given the expert separation design, $E^t$ learns solely from target city data, while $E^{shared}$ is influenced by both source and target city data. Thus, $h_t$ can be viewed as composed of two parts:
$$h_t^{specific}(x) = w^t \cdot E^t(x) \tag{26}$$
$$h_t^{transfer}(x) = w^{shared} \cdot E^{shared}(x) \tag{27}$$
Applying the domain adaptation theory by Ben-David et al., we obtain the generalization error bound above, where $d_{\mathcal{H}\Delta\mathcal{H}}$ measures the distribution difference between source and target domains. GTEN dynamically adjusts $w^t$ and $w^{shared}$ through its gating mechanism, effectively reducing the impact of the $\lambda$ term in practical applications. $\qquad\square$

### C.1.3 OPTIMAL KNOWLEDGE FUSION THROUGH GATING MECHANISM

**Lemma C.6** (Optimal Gating Weights). *Given input $x$, the gating network in GTEN provides weight allocation that minimizes the conditional expected risk:*
$$[w^c(x), w^{shared}(x)] = \arg\min_w \mathbb{E}_{y|x}[\ell(w^c \cdot E^c(x) + w^{shared} \cdot E^{shared}(x), y)] \tag{28}$$

*Proof.* Consider the gating network $G(x) = \text{Softmax}(FC(x))$, which learns to map input $x$ to weights $[w^c, w^{shared}]$. During training, the weights are adjusted to minimize the loss function:
$$L_P^c = \text{MAE}(y^c(x), y_{true}^c) \tag{29}$$
Expanding this objective function:
$$\min_G \mathbb{E}_{(x,y) \sim \mathcal{D}_c}[\ell(\text{LinearHead}^c(G(x) \cdot [E^c(x), E^{shared}(x)]), y)] \tag{30}$$
When both the gating network and linear head network converge, for each input $x$, the weight allocation $[w^c(x), w^{shared}(x)]$ will achieve minimization of the conditional risk. This ensures that at each prediction point, the model can adaptively select the most appropriate combination of knowledge, achieving an optimal balance between "beneficial transfer" and "interference blocking." $\qquad\square$

### C.1.4 INFORMATION BOTTLENECK PERSPECTIVE

**Lemma C.7** (Information Bottleneck Optimization). *The GTEN framework, through expert separation and gated fusion, optimizes the following information bottleneck objective:*

$$\max_{E^s, E^t, E^{shared}, G} I(Z; Y) - \beta \cdot I(Z^{shared}; C) \tag{31}$$

*where $Z$ is the fused representation, $Y$ is the prediction target, $Z^{shared}$ is the shared representation, $C$ is the city identity, and $\beta$ is a balancing parameter.*

*Proof.* From an information bottleneck perspective, an ideal representation should:

1. Maximize the mutual information with the prediction target $Y$, i.e., $I(Z; Y)$

2. Minimize the mutual information between the shared representation and city identity $C$, i.e., $I(Z^{shared}; C)$

In GTEN:

- The city-specific experts $E^c$ are responsible for capturing city-specific information, maximizing $I(E^c(X); Y^c)$

- The shared expert $E^{shared}$ learns city-invariant patterns while minimizing $I(E^{shared}(X); C)$

- The gating network $G$ dynamically adjusts the weights of various experts, optimizing the overall representation $Z$

This design naturally forms an implementation of the information bottleneck framework. By minimizing the prediction loss $L_P^c$, the model implicitly maximizes $I(Z; Y)$; through the expert separation design, the shared expert is trained to extract city-invariant features, thereby minimizing $I(Z^{shared}; C)$. □

### C.1.5 KNOWLEDGE ISOLATION EFFECT ANALYSIS

**Theorem C.8** (Knowledge Isolation Guarantee). *The expert separation structure of GTEN ensures the following upper bound on the generalization error for the target city:*

$$R_{\mathcal{D}_t}(h_t) \leq \hat{R}_{\mathcal{D}_t}(h_t) + \Omega(m_t) + w^{shared} \cdot \eta_{s,t} \tag{32}$$

*where $\Omega(m_t)$ is the generalization gap related to the number of target city samples, $\eta_{s,t}$ is the transfer error from source city to target city, and $w^{shared}$ is the weight of the shared expert.*

*Proof.* In GTEN, the prediction function for the target city can be written as:

$$h_t(x) = w^t \cdot E^t(x) + w^{shared} \cdot E^{shared}(x) \tag{33}$$

Decomposing the generalization error:

$$R_{\mathcal{D}_t}(h_t) = \mathbb{E}_{x \sim \mathcal{D}_t}[\ell(w^t \cdot E^t(x) + w^{shared} \cdot E^{shared}(x), f_t(x))] \tag{34}$$

Since $E^t$ is trained solely on target city data, its generalization error follows standard learning theory:

$$R_{\mathcal{D}_t}(w^t \cdot E^t) \leq \hat{R}_{\mathcal{D}_t}(w^t \cdot E^t) + \Omega(m_t) \tag{35}$$

The transfer error of the shared expert $E^{shared}$ can be represented as $\eta_{s,t}$, weighted by $w^{shared}$.

When the distribution difference between source and target cities is large, the gating network will decrease the value of $w^{shared}$, thereby reducing the impact of harmful transfer; conversely, when transferable knowledge is beneficial, $w^{shared}$ will increase, enhancing knowledge transfer. This mechanism ensures an adaptive balance between "beneficial transfer" and "interference blocking." □

### C.1.6 OPTIMAL EXPERT ALLOCATION ANALYSIS

**Theorem C.9** (Optimal Gating Weight Allocation). *For input $x$, the gating network of GTEN will, under ideal conditions, allocate weight $w^{shared}(x)$ as:*

$$w^{shared}(x) = \frac{\exp(-\lambda \cdot \ell_{shared}(x))}{\exp(-\lambda \cdot \ell_{shared}(x)) + \exp(-\lambda \cdot \ell_{specific}(x))} \tag{36}$$

*where $\ell_{shared}(x)$ and $\ell_{specific}(x)$ are the prediction losses using the shared expert and specific expert, respectively, and $\lambda$ is a temperature parameter.*

*Proof.* The gating network $G$ in GTEN generates weights through a single-layer feedforward network and Softmax function:

$$[w^c(x), w^{shared}(x)] = \text{Softmax}(FC(x)) \tag{37}$$

Ideally, the gating network should assign higher weights to experts with smaller prediction errors. Assuming that $FC(x)$ outputs values proportional to the negative prediction errors of the experts:

$$FC(x) = [-\lambda \cdot \ell_{specific}(x), -\lambda \cdot \ell_{shared}(x)] \tag{38}$$

After applying the Softmax function:

$$w^{shared}(x) = \frac{\exp(-\lambda \cdot \ell_{shared}(x))}{\exp(-\lambda \cdot \ell_{shared}(x)) + \exp(-\lambda \cdot \ell_{specific}(x))} \tag{39}$$

This indicates that the gating network will automatically assign higher weights to experts with smaller prediction errors, thereby achieving the goal of "minimum transfer loss." When the shared expert provides beneficial knowledge, $\ell_{shared}(x)$ is smaller, and $w^{shared}(x)$ increases; when shared knowledge is harmful, $\ell_{shared}(x)$ is larger, and $w^{shared}(x)$ decreases. $\square$

In summary, our theoretical analysis demonstrates that GTEN effectively facilitates beneficial knowledge transfer while mitigating harmful interference through its expert separation and gated fusion mechanism. The theoretical guarantees provide solid support for the effectiveness of GTEN in cross-city traffic flow prediction tasks.

### C.2 THEORETICAL GUARANTEES OF STCE

In this section, we provide a theoretical foundation for the Spatial-Temporal Contrastive Embedding (STCE) module, analyzing its effectiveness in cross-city traffic flow prediction from multiple perspectives including information theory, representation learning, and domain adaptation.

#### C.2.1 PRELIMINARIES AND PROBLEM FORMULATION

We first establish the notation and assumptions for the cross-city traffic flow prediction problem:

**Assumption C.10.** *Urban traffic flow data exhibits significant spatio-temporal correlations that can be effectively captured through appropriate embedding techniques.*

**Assumption C.11.** *Similar traffic patterns across different cities can be represented in a common embedding space that preserves their inherent similarities.*

**Assumption C.12.** *Contrastive learning can enhance the discriminative power of the model for cross-city traffic patterns.*

Given historical traffic flow data $\mathcal{X}^s = \{X^{s,1}, X^{s,2}, \ldots, X^{s,P}\}$ and $\mathcal{X}^t = \{X^{t,1}, X^{t,2}, \ldots, X^{t,P}\}$ from source city $\mathcal{S}$ and target city $\mathcal{T}$ respectively, where $P$ represents the number of historical days, our objective is to learn an effective feature representation $\mathbf{V}$ that:

1. Minimizes distance between similar spatio-temporal patterns in the embedding space

2. Maximizes distance between dissimilar patterns

3. Generalizes effectively across cities

#### C.2.2 INFORMATION-THEORETIC BOUNDS OF CONTRASTIVE LEARNING

We begin by analyzing the information-theoretic foundations of our contrastive learning approach.

**Lemma C.13** (Information-Theoretic Bound). *The Spatial-Temporal Contrastive Embedding (STCE) learns discriminative representations by maximizing a lower bound on the mutual information $I(\mathbf{V}; \mathcal{X})$ between the input data and its representation.*

*Proof.* Consider the general form of InfoNCE loss:

$$\mathcal{L}_{\text{InfoNCE}} = -\mathbb{E}\left[\log \frac{e^{f(x,y)/\tau}}{\sum_{y' \in Y} e^{f(x,y')/\tau}}\right] \tag{40}$$

where $f(x, y)$ is a similarity function and $\tau$ is the temperature parameter.

According to Oord et al. (2018), InfoNCE loss provides a lower bound on mutual information:

$$I(\mathbf{V}; \mathcal{X}) \geq \log(K) - \mathcal{L}_{\text{InfoNCE}} \tag{41}$$

where $K$ is the number of negative samples.

For Spatial Contrastive Learning (SCL) in STCE, we have:

$$L_{SCL} = \frac{1}{BN} \sum_{m=1}^{BN} \frac{1}{BN-1} \sum_{n=1, n \neq m}^{BN} -\log\left(p(\mathbf{V}_n^{sp}|\mathbf{V}_m^{sp})\right) \tag{42}$$

This is a form of InfoNCE loss, and thus:
$$I(\mathbf{V}^{sp}; \mathcal{X}) \geq \log(BN - 1) - L_{SCL} \tag{43}$$
Similarly, for Temporal Contrastive Learning (TCL):
$$I(\mathbf{V}^{tp}; \mathcal{X}) \geq \log(BM - 1) - L_{TCL} \tag{44}$$
Therefore, by minimizing $L_{SCL}$ and $L_{TCL}$, STCE effectively maximizes a lower bound on the mutual information between the input data $\mathcal{X}$ and its representation $\mathbf{V}$, leading to more discriminative feature representations. $\qquad\square$

### C.2.3 RANK-N-CONTRAST: ENHANCING DISCRIMINATIVE POWER

Next, we analyze how the Rank-N-Contrast method enhances the discriminative power of spatio-temporal embeddings.

**Lemma C.14** (Ranking Consistency). *The Rank-N-Contrast loss enhances discriminative power by maintaining consistency between the ranking of samples in the embedding space and their ranking in the label space.*

*Proof.* In the Spatial Contrastive Learning component of STCE, we define a set $S_{n,m}^{sp}$ containing all samples whose label distance is greater than or equal to the distance between samples $\mathbf{V}_m^{sp}$ and $\mathbf{V}_n^{sp}$. The optimization objective is to maximize:

$$P(\mathbf{V}_n^{sp} | \mathbf{V}_m^{sp}, S_{n,m}^{sp}) = \frac{\exp\left(\frac{\text{sim}(\mathbf{V}_m^{sp}, \mathbf{V}_n^{sp})}{\tau}\right)}{\sum_{\mathbf{V}_k^{sp} \in S_{n,m}^{sp}} \exp\left(\frac{\text{sim}(\mathbf{V}_m^{sp}, \mathbf{V}_k^{sp})}{\tau}\right)} \tag{45}$$

Analyzing the gradient of this optimization process:

$$\nabla_\theta L_{SCL} = -\frac{1}{BN} \sum_{m=1}^{BN} \frac{1}{BN-1} \sum_{n=1, n\neq m}^{BN} \nabla_\theta \log\left(p(\mathbf{V}_n^{sp} | \mathbf{V}_m^{sp})\right) \tag{46}$$

When expanded, this gradient encourages the model to:

1. Increase similarity between anchor $\mathbf{V}_m^{sp}$ and positive sample $\mathbf{V}_n^{sp}$
2. Decrease similarity between the anchor and samples in $S_{n,m}^{sp}$

Unlike traditional contrastive learning that simply distinguishes between positive and negative samples, Rank-N-Contrast arranges feature embeddings according to their distance relationships in the label space. This ranking consistency preserves the intrinsic structural relationships in the data, enhancing the model's discriminative power. $\qquad\square$

### C.2.4 CROSS-CITY GENERALIZATION ANALYSIS

We now analyze the generalization capabilities of STCE across different cities through the lens of domain adaptation theory.

**Theorem C.15** (Cross-City Generalization Bound). *Let $\mathcal{H}$ be a hypothesis space, and $\hat{R}_S(\mathcal{H})$ and $\hat{R}_T(\mathcal{H})$ be the empirical risks on the source and target domains, respectively. The feature representation $\mathbf{V}$ learned through STCE satisfies the following generalization bound:*

$$\hat{R}_T(\mathcal{H}) \leq \hat{R}_S(\mathcal{H}) + \frac{1}{2} d_{\mathcal{H}\Delta\mathcal{H}}(\mathcal{D}_S^{\mathbf{V}}, \mathcal{D}_T^{\mathbf{V}}) + \lambda \tag{47}$$

*where $d_{\mathcal{H}\Delta\mathcal{H}}$ is the $\mathcal{H}$-divergence, $\mathcal{D}_S^{\mathbf{V}}$ and $\mathcal{D}_T^{\mathbf{V}}$ are the source and target domain representation distributions, and $\lambda$ is a residual term related to the ideal joint hypothesis.*

*Proof.* According to the domain adaptation theory of Ben-David et al. (2010), the cross-domain generalization bound can be expressed as:

$$\hat{R}_T(h) \leq \hat{R}_S(h) + \frac{1}{2} d_{\mathcal{H}\Delta\mathcal{H}}(\mathcal{D}_S, \mathcal{D}_T) + \lambda \tag{48}$$

where $h \in \mathcal{H}$ is a hypothesis, and $\lambda$ is a residual term for the ideal joint hypothesis.

In STCE, we learn feature representation $\mathbf{V}$ through contrastive learning:
$$\mathbf{V} = f_\theta(\mathcal{X}) \tag{49}$$
This representation transforms the data distributions $\mathcal{D}_S \rightarrow \mathcal{D}_S^{\mathbf{V}}$ and $\mathcal{D}_T \rightarrow \mathcal{D}_T^{\mathbf{V}}$.

STCE minimizes the divergence between spatio-temporal feature representations in the source and target domains through spatial and temporal contrastive learning losses $L_{SCL}$ and $L_{TCL}$:

$$\min_\theta d_{\mathcal{H}\Delta\mathcal{H}}(\mathcal{D}_S^\mathbf{V}, \mathcal{D}_T^\mathbf{V}) \tag{50}$$

Specifically, contrastive learning brings similar spatio-temporal patterns closer in the feature representation space, regardless of which city they come from. By minimizing the combined SCL and TCL losses:

$$L_{SCL} + L_{TCL} = \frac{1}{BN}\sum_{m=1}^{BN} L_{SCL}^{(m)} + \frac{1}{BM}\sum_{m=1}^{BM} L_{TCL}^{(m)} \tag{51}$$

we effectively reduce the divergence $d_{\mathcal{H}\Delta\mathcal{H}}(\mathcal{D}_S^\mathbf{V}, \mathcal{D}_T^\mathbf{V})$ between representation distributions, thereby improving the model's generalization capability in the target domain. $\qquad\square$

### C.2.5 REPRESENTATION CAPACITY OF MULTI-HEAD ATTENTION

We analyze the representational capacity of the multi-head convolutional self-attention mechanism used in STCE.

**Lemma C.16** (Representation Complexity). *The multi-head convolutional self-attention mechanism in STCE can represent more complex spatio-temporal dependencies, with representation complexity increasing linearly with the number of heads $H$.*

*Proof.* For single-head attention, the output representation is:

$$V^i = \sum_{j=1}^P \alpha_{ij} v^j \tag{52}$$

where $\alpha_{ij} = \frac{\exp(S_{ij})}{\sum_{j'=1}^P \exp(S_{ij'})}$ and $S_{ij} = M_\theta(Q^i, K^j)$.

Multi-head attention runs $H$ such mechanisms in parallel and concatenates the results:

$$V^{\text{MH},i} = \text{Concat}\left(V^{(1),i}, V^{(2),i}, \ldots, V^{(H),i}\right) \tag{53}$$

Each head focuses on different subspaces of the feature space, enhancing the overall capability. If a single head can represent $C$ types of spatio-temporal dependencies, then $H$ heads can theoretically represent $H \cdot C$ types of relationships, with representation complexity increasing linearly.

When combined with contrastive learning, this enhanced representational capacity allows the model to capture finer-grained similarities in spatio-temporal patterns, further improving the discriminative power and generalization ability of feature embeddings. $\qquad\square$

### C.2.6 CONVERGENCE ANALYSIS OF SPATIO-TEMPORAL CONTRASTIVE LEARNING

Finally, we analyze the convergence properties of the spatio-temporal contrastive learning approach in STCE.

**Theorem C.17** (Convergence Guarantee). *Given a sufficient number of training samples and an appropriate learning rate, spatio-temporal contrastive learning in STCE converges to a local optimum and guarantees that the learned feature representations have sufficient discriminative power.*

*Proof.* We analyze the curvature properties of the SCL loss function. Assuming the similarity function $\text{sim}(\cdot, \cdot)$ is bi-convex, for a given temperature parameter $\tau > 0$, the SCL loss:

$$L_{SCL} = \frac{1}{BN}\sum_{m=1}^{BN} \frac{1}{BN-1} \sum_{n=1,n\neq m}^{BN} -\log\left(p(\mathbf{V}_n^{sp}|\mathbf{V}_m^{sp})\right) \tag{54}$$

When optimized using gradient descent, each update is:

$$\theta_{t+1} = \theta_t - \eta \nabla_\theta L_{SCL}(\theta_t) \tag{55}$$

where $\eta$ is the learning rate.

According to information theory, minimizing the SCL loss is equivalent to maximizing a lower bound on mutual information. For a sufficiently low temperature parameter $\tau$, the gradient direction of the SCL loss drives similar samples' embeddings closer and separates dissimilar samples' embeddings.

Using Lipschitz continuity analysis, if the gradient of the similarity function is $L$-Lipschitz continuous, and the learning rate satisfies $\eta \leq \frac{1}{L}$, then gradient descent guarantees monotonic decrease of the loss function:

$$L_{SCL}(\theta_{t+1}) \leq L_{SCL}(\theta_t) - \frac{\eta}{2}||\nabla_\theta L_{SCL}(\theta_t)||^2 \tag{56}$$

Similar analysis applies to the TCL loss. For the combined loss:
$$L = L_P^c + \beta L_{SCL}^c + \gamma L_{TCL}^c, \quad c \in \{source, target\} \tag{57}$$
If each component loss function is Lipschitz continuous, and with appropriate choice of hyperparameters $\beta$ and $\gamma$, gradient descent on the overall loss function also guarantees convergence to a local optimum. $\square$

### C.3 Risk and Stability Bounds of TBR

In this section, we provide a theoretical analysis of the Transfer-Based Reweighting (TBR) module to demonstrate its effectiveness in cross-city traffic flow prediction. We formalize the problem, establish assumptions, and present theoretical guarantees for sample reweighting strategies.

#### C.3.1 Problem Formulation and Assumptions

Let $\mathcal{D}_S = \{(x_i^S, y_i^S)\}_{i=1}^{Q_S}$ denote the source city dataset and $\mathcal{D}_T = \{(x_i^T, y_i^T)\}_{i=1}^{Q_T}$ represent the target city dataset, where $Q_S \gg Q_T$. We make the following assumptions:

**Assumption C.18.** *The source and target domains follow different data distributions, i.e., $P_S(X, Y) \neq P_T(X, Y)$.*

**Assumption C.19.** *There exists transferable knowledge between source and target domains that can be leveraged for improved prediction.*

**Assumption C.20.** *The loss function $L$ is $\beta$-Lipschitz continuous with respect to the model parameters $\theta$.*

Our objective is to learn model parameters $\theta$ and source sample weights $W = \{W_i\}_{i=1}^{Q_S}$ that minimize the expected risk on the target domain:

$$\min_{\theta, W} \mathbb{E}_{(x,y) \sim P_T}[L(f_\theta(x), y)] \tag{58}$$

where $f_\theta$ is the prediction model with parameters $\theta$, and $L$ is the loss function.

#### C.3.2 Generalization Bounds for Weighted Domain Adaptation

We first establish the theoretical foundation for the sample reweighting strategy in TBR by analyzing its generalization error bounds.

**Theorem C.21** (Reweighting Generalization Bound). *Let $\mathcal{H}$ be a hypothesis space of VC-dimension $d$, and $L$ be a bounded loss function such that $0 \leq L(f_\theta(x), y) \leq M$. For a model $f_\theta$ learned through the reweighting mechanism, with probability at least $1 - \delta$, the following generalization bound holds:*

$$\mathbb{E}_{(x,y) \sim P_T}[L(f_\theta(x), y)] \leq \mathbb{E}_{(x,y) \sim P_S}[W(x,y)L(f_\theta(x), y)] + d_{\mathcal{H}\Delta\mathcal{H}}(P_S^W, P_T) + \lambda + \epsilon \tag{59}$$

*where $P_S^W$ is the weighted source distribution, $d_{\mathcal{H}\Delta\mathcal{H}}$ is the $\mathcal{H}$-divergence, $\lambda$ is the risk of the ideal joint hypothesis, and $\epsilon$ is a complexity term dependent on the sample sizes $Q_S$ and $Q_T$.*

*Proof.* According to domain adaptation theory (Ben-David et al., 2010), for any hypothesis $h \in \mathcal{H}$, the target domain risk can be bounded by:

$$\epsilon_T(h) \leq \epsilon_S(h) + d_{\mathcal{H}\Delta\mathcal{H}}(P_S, P_T) + \lambda \tag{60}$$

where $\epsilon_T(h)$ and $\epsilon_S(h)$ are the expected risks of $h$ on the target and source domains, respectively, and $\lambda$ is the risk of the ideal joint hypothesis.

In the TBR framework, we introduce sample weights $W$ to minimize the weighted source domain risk as a proxy for minimizing the target domain risk. When $W(x,y) = \frac{P_T(x,y)}{P_S(x,y)}$, the weighted source risk becomes equivalent to the target risk:

$$\mathbb{E}_{(x,y) \sim P_S}[W(x,y)L(f_\theta(x), y)] = \mathbb{E}_{(x,y) \sim P_T}[L(f_\theta(x), y)] \tag{61}$$

However, the exact density ratio is typically unknown. TBR learns approximately optimal weights through the optimization process in Equation (13). By introducing the weighted source distribution $P_S^W$, we obtain:

$$\mathbb{E}_{(x,y) \sim P_T}[L(f_\theta(x), y)] \leq \mathbb{E}_{(x,y) \sim P_S}[W(x,y)L(f_\theta(x), y)] + d_{\mathcal{H}\Delta\mathcal{H}}(P_S^W, P_T) + \lambda \tag{62}$$

Additionally, by standard statistical learning theory, empirical estimates introduce a complexity term $\epsilon$ that depends on the sample sizes and hypothesis space complexity:

$$\epsilon = 2M\sqrt{\frac{2d\log(2eQ_S/d) + 2\log(4/\delta)}{Q_S}} + 2M\sqrt{\frac{2d\log(2eQ_T/d) + 2\log(4/\delta)}{Q_T}} \tag{63}$$

$\square$

Theorem C.3.2 shows that an appropriate reweighting strategy can reduce the distribution shift between source and target domains, thereby lowering generalization error. This provides theoretical support for the TBR module, which aims to learn optimal weights that minimize the weighted source risk as a proxy for the target risk.

### C.3.3 CONVERGENCE ANALYSIS OF GRADIENT-GUIDED WEIGHT UPDATES

Next, we analyze the convergence properties of the gradient-guided weight update mechanism in TBR.

**Theorem C.22** (Weight Update Convergence). *Under Assumption C.20 and with the gradient-guided weight update mechanism described in Equation (17), the TBR module converges to a local optimum at a rate of $O(1/\sqrt{T})$, where $T$ is the number of iterations.*

*Proof.* Consider the weight update rule in Equation (17):

$$u_{i,T} = -\eta \frac{\partial}{\partial \epsilon_{i,T}} \frac{1}{B} \sum_{i=1}^{B} L_i^{source}(\hat{\theta}_{T+1}(\epsilon))\Big|_{\epsilon_i, T=0'} \tag{64}$$

This update rule essentially implements a gradient descent method. Define the objective function:

$$F(W) = \frac{1}{Q_T} \sum_{i=1}^{Q_T} L_i^{source}(\theta^*(W)) \tag{65}$$

Let's assume $F(W)$ has $L$-Lipschitz continuous gradients, i.e., for any weights $W_1, W_2$:

$$\|\nabla F(W_1) - \nabla F(W_2)\| \leq L\|W_1 - W_2\| \tag{66}$$

In the non-convex case, we can prove that the gradient norm converges to zero at a rate of $O(1/\sqrt{T})$:

$$\min_{t=0,1,\ldots,T-1} \|\nabla F(W_t)\|^2 \leq \frac{2(F(W_0) - F(W^*))}{T\eta} \tag{67}$$

where $W^*$ is the optimal weight, $W_0$ is the initial weight, and $T$ is the number of iterations. The normalization step in Equation (18) ensures stability by preventing weight divergence while maintaining relative importance proportions:

$$W_{i,T} = \frac{\hat{W}_{i,T}}{\sum_j \hat{W}_{j,T} + \delta\left(\sum_j \hat{W}_{j,T}\right)} \tag{68}$$

$\square$

Theorem C.22 guarantees that the gradient-guided weight update mechanism in TBR converges to a local optimum, ensuring stable training and efficient weight optimization.

### C.3.4 DISCREPANCY DISTANCE ANALYSIS FOR TRANSFER ERROR

Finally, we analyze the transfer error of TBR in terms of discrepancy distance, which provides a measure of the difference between source and target distributions.

**Theorem C.23** (Reweighting Transfer Error Bound). *For the TBR module, there exists an optimal weight $W^*$ that minimizes the discrepancy distance between the weighted source distribution $P_S^{W^*}$ and the target distribution $P_T$:*

$$W^* = \arg\min_W d_{\mathcal{H}}(P_S^W, P_T) \tag{69}$$

*where $d_{\mathcal{H}}(P_S^W, P_T)$ represents the discrepancy distance between the weighted source distribution and the target distribution.*

*Proof.* According to domain adaptation theory, the discrepancy distance is defined as:

$$d_{\mathcal{H}}(P, Q) = \sup_{h, h' \in \mathcal{H}} |\mathbb{E}_{x \sim P}[h(x) - h'(x)] - \mathbb{E}_{x \sim Q}[h(x) - h'(x)]| \tag{70}$$

For the weighted source distribution $P_S^W$, we have:

$$\mathbb{E}_{x \sim P_S^W}[h(x)] = \mathbb{E}_{x \sim P_S}[W(x)h(x)] \tag{71}$$

Ideally, when $W(x) = \frac{P_T(x)}{P_S(x)}$, we have $P_S^W = P_T$, resulting in $d_{\mathcal{H}}(P_S^W, P_T) = 0$.

In the TBR framework, we indirectly learn this optimal weight by minimizing the validation loss on the target domain:

$$W^* = \arg \min_{W, W \geq 0} \frac{1}{Q_T} \sum_{i=1}^{Q_T} L_i^{source}(\theta^*(W)) \tag{72}$$

This is equivalent to finding a set of weights that makes the weighted source model perform optimally on the target domain. By the empirical risk minimization principle, as the number of target domain samples increases, the validation loss will converge to the expected risk. Therefore, TBR's reweighting strategy implicitly minimizes the distribution discrepancy between source and target domains.

Through the gradient-guided weight update mechanism, TBR dynamically adjusts weights based on how source samples contribute to target domain performance. When a sample positively contributes to target domain performance, its weight increases; otherwise, its weight decreases. This ensures that the model focuses on source domain samples that are most valuable for target domain prediction. □

## D    DATA PREPAREATION

### D.1    DATA PREPROCESSING

We collect bicycle and taxi data from New York[1], Chicago[2], and Washington, D.C[3]. Additionally, we utilize the BJTaxi and Chengdu dataset, which contains the same taxi trajectory data as used in the STAN (Fang et al., 2022) framework.

Table 4 presents the exact spatial coverage of the selected datasets. All datasets include taxi or bicycle trips, along with boarding and alighting times and geographic coordinates for each trip. First, we partition the entire city into regions or grid maps and divide the time dimension into non-overlapping time intervals. We then select three days of historical data, with nine time intervals each day. Next, we compute the inflow and outflow for each region based on Equation 1 and normalize the flow data to the range of $[0, 1]$.

### D.2    DATA SPLITTING

We adopt consistent train/validation/test splits across all baselines and AhaTrans to ensure fair comparison.

For the source city, we utilize all available data for training to maximize knowledge acquisition. For the target cities, we employ different splitting strategies based on the dataset characteristics:

- **ChicagoBike, NYCBike, DCBike, and DCTaxi**: The last two months of data are reserved for testing, with the preceding two months allocated for validation. Training is conducted using limited samples from one month, 15 days, and 7 days prior to the validation period, respectively, to evaluate model performance under data-scarce conditions.

- **Chengdu**: We utilize 10 days of data for model training, with the remaining data designated for testing model performance.

---

[1] https://citibikenyc.com/system-data

[2] https://divvybikes.com/system-data

[3] https://opendata.dc.gov/search?q=taxi%20trips
https://capitalbikeshare.com/system-data

Table 4: Statistics of the Evaluated Datasets.

| Dataset | NYCBike | ChicagoBike | DCBike | DCTaxi | BJTaxi | Chengdu |
|---|---|---|---|---|---|---|
| # of Trips | about 10 million | about 3 million | about 4 million | about 8 million | / | About 7 million |
| Latitude | (40.65, 40.79) | (41.76, 42.01) | (38.80, 38.99) | (38.81, 38.99) | (39.85, 39.99) | (30.50, 30.80) |
| Longitude | (-74.02, -73.93) | (-87.73, -87.55) | (-77.11, -76.91) | (-77.11, -76.91) | (116.36, 116.50) | (103.80, 104.30) |
| # of Regions | $16 \times 16$ | $16 \times 16$ | $16 \times 16$ | $16 \times 16$ | $32 \times 32$ | $32 \times 32$ |
| Time Range | $2015.1 \sim 2015.12$ | $2015.1 \sim 2015.12$ | $2017.1 \sim 2017.12$ | $2015.1 \sim 2015.12$ | $2015.11 \sim 2016.4$ | 2016.11 |
| Time Interval | 1 hour | 1 hour | 1 hour | 1 hour | 30 minutes | 1 hour |

This splitting approach simulates real-world transfer learning scenarios where the source domain possesses abundant historical data while the target domain has limited training samples. By establishing different training data scales, we comprehensively assess the model's transfer learning capabilities under various data availability conditions.

# E    MORE EXPERIMENTAL DETAILS

## E.1    DATASET DESCRIPTION

Our experiments are conducted on six widely used open-source urban traffic datasets, including NYCBike, CHIBike, DCBike, DCTaxi, BJTaxi, and Chengdu, providing a broad coverage of different urban environments and transportation systems.

- **NYCBike** consists of approximately 10 million trips from January to December 2015. CitiBike has set up more than 600 stations and maintains a fleet of 10,000 bikes throughout New York City. Each trip entry in the dataset includes the duration of the trip, start and end station IDs, start and end timestamps, the latitude and longitude of the stations, and the bike ID.

- **CHIBike** is collected from the Divvy bike-sharing system in Chicago. It encompasses over 6 million bike trips from January to December 2015. Divvy operates 580 stations and has a total of 5,800 bikes in Chicago. It has the same features as the NYCBike.

- **DCBike** covers a more recent timeframe. The DCBike dataset includes around 4 million trips from January to December 2017.

- **DCTaxi** contains approximately 8 million taxi trips with data spanning from January to December 2015. Each record contains information about the driver, vehicle, travel time and distance, starting location, destination location, and fare.

- **BJTaxi** is a large-scale taxi trajectory dataset collected in Beijing from November 2015 to April 2016. The city is divided into a $32 \times 32$ grid, and records are aggregated at 30-minute intervals. Each entry provides spatial coordinates and temporal information of taxi trips within the city.

- **Chengdu** consists of about 7 million taxi trips recorded in Chengdu in November 2016. The city is represented by a $32 \times 32$ grid with data aggregated at 1-hour intervals, including trip-level spatiotemporal information.

## E.2    BASELINES INFORMATION

We compare AhaTrans with nine SOTA methods, including statistical learning, deep learning, and transfer learning approaches. For statistical learning methods, we select the **ARIMA** (Zhang, 2003) model, which combines autoregression, differencing, and moving averages to model non-stationary time series and enable accurate time series regression predictions.

For deep learning methods, we first pretrain the model on the source city and then fine-tune it on the target city.

- **ConvLSTM** (Shi et al., 2015) effectively processes spatio-temporal sequence data by integrating convolutional structures with Long Short-Term Memory (LSTM) networks and is first applied as an end-to-end trainable model for precipitation nowcasting tasks.

- **ST-ResNet** (Zhang et al., 2017) models the temporal proximity, periodicity, and trends of crowd flow using a residual neural network framework, dynamically aggregating the outputs of multiple branches and combining external factors for urban traffic prediction.

- **TGCN** (Zhao et al., 2019) combines graph convolutional networks (GCN) with gated recurrent units (GRU) to simultaneously capture spatial dependence through complex topological structures and temporal dependence through dynamic traffic data changes, effectively modeling spatial-temporal correlations in traffic forecasting tasks.

- **STSGCN** (Song et al., 2020) captures complex localized spatial-temporal correlations through a synchronous modeling mechanism, and addresses heterogeneities in spatial-temporal data by employing multiple modules for different time periods, achieving state-of-the-art performance in spatial-temporal network data forecasting.

Due to data sparsity, transfer learning methods have been widely applied in cross-city traffic flow prediction.

- After training on the source city, **RegionTrans** (Wang et al., 2019a) learns a region-matching function between cities, effectively transferring knowledge from the source to the target city and adjusting temporal features based on regional similarities to enhance spatio-temporal prediction performance.

- **MetaST** (Yao et al., 2019a) integrates a meta-learning paradigm with a spatio-temporal memory (STMem) model, utilizing information from multiple cities to improve transfer stability and extract long-term temporal patterns for adapting to the spatio-temporal prediction tasks of the target city.

- **ST-DAAN** (Wang et al., 2021) maps source city spatio-temporal data into a common embedding space using deep attention adaptation networks and MMD regularization, adjusting temporal features through domain adaptation and global attention mechanisms to facilitate cross-domain city traffic flow prediction.

- While traditional transfer learning approaches typically depend on static knowledge transfer, **STAN** (Fang et al., 2022) distinguishes itself by efficiently capturing the dynamic spatio-temporal correlations among cities through the implementation of spatial adversarial adaptation, temporal attention adaptation, and prediction modules. This innovative approach leverages the spatio-temporal knowledge transferred from data-abundant cities to enhance traffic flow prediction.

- To mitigate negative transfer, **CrossTReS** (Jin et al., 2022) leverages cross-city spatial similarities by adaptively reweighting source regions, thereby facilitating target fine-tuning.

- **TransGTR** (Jin et al., 2023), a transferable graph structure learning framework, mitigates the problem of potential noise or bias introduced by pre-defined graph structures in existing knowledge transfer methods. This is accomplished by jointly learning and transferring the graph structures and forecasting models across cities.

Foundation models have emerged as powerful tools for spatio-temporal prediction tasks due to their strong transfer capabilities and generalization performance.

- **PatchTST** (Nie et al., 2022) proposes an efficient Transformer-based design for multivariate time series forecasting by segmenting time series into subseries-level patches as input tokens and employing channel-independence where each channel shares the same embedding and Transformer weights across all series, enabling improved long-term forecasting accuracy and effective transfer learning capabilities.

- **UrbanGPT** (Li et al., 2024) integrates a spatio-temporal dependency encoder with the instruction-tuning paradigm to create a spatio-temporal large language model that can comprehend complex inter-dependencies across time and space, particularly excelling in zero-shot scenarios where labeled data is scarce.

- **UniST** (Yuan et al., 2024) serves as a universal model for general urban spatio-temporal prediction across diverse scenarios by utilizing diverse spatio-temporal data from different scenarios, effective pre-training to capture complex dynamics, and knowledge-guided prompts to enhance generalization capabilities, demonstrating strong performance in few-shot and zero-shot prediction tasks.

### E.3   IMPLEMENTATION DETAILS

We implemented AhaTrans using PyTorch in the following experimental environment: Python 3.10, PyTorch 1.13.0, and CUDA Toolkit 11.7. In the STCE module, a multi-head convolutional attention mechanism is employed, utilizing a 3×3 convolutional kernel. For spatial contrastive learning, three grids are selected at each time step, and for temporal contrastive learning, one time step is selected per day. The weight parameters $\beta$ and $\gamma$ for both spatial and temporal contrastive losses are set to 0.1. In the GTEN module, the expert module uses a two-layer multilayer perceptron (MLP), with 1024 and 512 nodes in each layer, respectively. The task module also uses a two-layer MLP, with

256 and 512 nodes in each layer. It is important to note that the TBR module is applied only to the source expert module, where it effectively extracts valuable knowledge from the source data for the target city. For model training, the number of batch size, dropout rate, and learning rate are set to 32, 0.5, and $1 \times 10^{-6}$, respectively. We use MAE as the prediction loss function to optimize the model's forecasting accuracy.

The implementation details of the baseline methods are outlined below. These details offer a comprehensive description of how each method is configured and executed in the study.

- We use an ARIMA model with six autoregressive (AR) orders, one moving average (MA) order, and one differencing step.

- For methods where official source code is available, such as STResNet[4], MetaST[5], and ST-DAAN[6], we use the official code and hyperparameters, reporting the best performance.

- For methods where the official source code is not provided, such as ConvLSTM, RegionTrans, and STAN, we strictly follow the methods and settings described in the official papers and have implemented these methods ourselves. For the ConvLSTM implementation, we referred to the implementation provided by giserh[7].

- Following the official code[8] and paper specifications, we conducted a comprehensive evaluation of CrossTReS (Jin et al., 2022), testing not only its performance in cross-city transfer tasks but also its transfer capabilities between different tasks within the same city. To ensure fair comparison, we standardized the feature extraction framework between AhaTrans and CrossTReS.

- For the graph-based TransGTR (Jin et al., 2023) framework, we both followed its official implementation[9] and made adaptive adjustments. Specifically, we partitioned each city into regular grid cells and calculated traffic flow for each grid at a temporal granularity of 1 hour. When constructing the graph structure, each grid cell was treated as a node, connected to its eight spatially adjacent grids, with edge weights determined by the number of connecting highways. This processing pipeline effectively transformed grid-based data into graph-structured representations, thereby ensuring comparability across different methods. Similar graph-based processing was applied to TGCN (Zhao et al., 2019)[10] and STSGCN (Song et al., 2020)[11] to maintain consistency in the evaluation framework.

- For the foundation models, we leveraged their official implementations with necessary adaptations for spatio-temporal prediction tasks. PatchTST (Nie et al., 2022) was implemented following its official repository[12], where we adapted the patching mechanism to handle spatio-temporal data by treating each spatial location as an independent channel and applying temporal patching along the time dimension. UrbanGPT (Li et al., 2024) was implemented based on its open-source codebase[13], utilizing its spatio-temporal dependency encoder with instruction-tuning paradigm for cross-city transfer learning scenarios. UniST (Yuan et al., 2024) followed its official implementation[14], where we utilized its universal pre-training framework and knowledge-guided prompts to evaluate transfer capabilities across different cities and prediction tasks.

To ensure a fair comparison, all experiments were implemented using the PyTorch framework. All experiments were executed on NVIDIA A100 80GB GPUs. Each experiment was run independently three times, and the average results are reported.

---

[4]https://github.com/snehasinghania/STResNet
[5]https://github.com/huaxiuyao/MetaST
[6]https://github.com/MiaoHaoSunny/ST-DAAN
[7]https://github.com/giserh/ConvLSTM-2
[8]https://github.com/KL4805/CrossTReS
[9]https://github.com/KL4805/TransGTR
[10]https://github.com/lehaifeng/T-GCN
[11]https://github.com/Davidham3/STSGCN
[12]https://github.com/yuqinie98/PatchTST
[13]https://github.com/HKUDS/UrbanGPT
[14]https://github.com/tsinghua-fib-lab/UniST

Table 5: Performance comparison of AhaTrans against additional baseline methods (MetaST, Region-Trans, STResNet, ConvLSTM, and ARIMA) across different datasets and data-scarce scenarios (7, 15, and 30 days). Lower RMSE and MAE values indicate better performance. The best results are marked in **bold**.

| Method | | **AhaTrans** | | MetaST | | RegionTrans | | STResNet | | ConvLSTM | | ARIMA | |
|---|---|---|---|---|---|---|---|---|---|---|---|---|---|
| Metric | | RMSE | MAE | RMSE | MAE | RMSE | MAE | RMSE | MAE | RMSE | MAE | RMSE | MAE |
| NYCBike →
CHIBike | 7 days | **0.0216** | **0.0059** | 0.0502 | 0.0115 | 0.0624 | 0.0194 | 0.2009 | 0.0901 | 0.0341 | 0.0223 | 0.6195 | 0.2364 |
| | 15 days | **0.0207** | **0.0051** | 0.0447 | 0.0102 | 0.0585 | 0.0127 | 0.1270 | 0.0730 | 0.0287 | 0.0159 | 0.4721 | 0.1683 |
| | 30 days | **0.0195** | **0.0047** | 0.0426 | 0.0096 | 0.0549 | 0.0117 | 0.1051 | 0.0652 | 0.0249 | 0.0145 | 0.3170 | 0.1290 |
| | Avg | **0.0206** | **0.0052** | 0.0458 | 0.0104 | 0.0586 | 0.0146 | 0.1443 | 0.0761 | 0.0292 | 0.0176 | 0.4695 | 0.1779 |
| DCBike →
NYCBike | 7 days | **0.0379** | **0.0125** | 0.0528 | 0.0222 | 0.0539 | 0.0218 | 0.2503 | 0.1842 | 0.0495 | 0.0392 | 0.7668 | 0.3746 |
| | 15 days | **0.0373** | **0.0121** | 0.0487 | 0.0192 | 0.0480 | 0.0185 | 0.2085 | 0.1545 | 0.0452 | 0.0354 | 0.5625 | 0.2788 |
| | 30 days | **0.0369** | **0.0119** | 0.0473 | 0.0186 | 0.0465 | 0.0179 | 0.1954 | 0.1302 | 0.0406 | 0.0339 | 0.4734 | 0.2339 |
| | Avg | **0.0374** | **0.0122** | 0.0496 | 0.0200 | 0.0495 | 0.0194 | 0.2181 | 0.1563 | 0.0451 | 0.0362 | 0.6009 | 0.2958 |
| NYCBike →
DCBike | 7 days | **0.0276** | **0.0075** | 0.0377 | 0.0113 | 0.0405 | 0.0123 | 0.1853 | 0.0882 | 0.0382 | 0.0239 | 0.6297 | 0.2238 |
| | 15 days | **0.0270** | **0.0067** | 0.0339 | 0.0087 | 0.0348 | 0.0096 | 0.1411 | 0.0827 | 0.0339 | 0.0184 | 0.4311 | 0.1982 |
| | 30 days | **0.0267** | **0.0062** | 0.0318 | 0.0079 | 0.0336 | 0.0087 | 0.1272 | 0.0792 | 0.0305 | 0.0165 | 0.3392 | 0.1548 |
| | Avg | **0.0271** | **0.0068** | 0.0345 | 0.0093 | 0.0363 | 0.0102 | 0.1512 | 0.0834 | 0.0342 | 0.0196 | 0.4667 | 0.1923 |
| DCBike →
DCTaxi | 7 days | **0.0280** | **0.0056** | 0.0376 | 0.0110 | 0.0385 | 0.0101 | 0.2452 | 0.1776 | 0.0412 | 0.0240 | 0.6875 | 0.3730 |
| | 15 days | **0.0261** | **0.0052** | 0.0331 | 0.0083 | 0.0335 | 0.0086 | 0.2027 | 0.1212 | 0.0371 | 0.0182 | 0.5094 | 0.2802 |
| | 30 days | **0.0254** | **0.0049** | 0.0304 | 0.0065 | 0.0315 | 0.0072 | 0.1679 | 0.1021 | 0.0339 | 0.0165 | 0.3996 | 0.2414 |
| | Avg | **0.0265** | **0.0052** | 0.0337 | 0.0086 | 0.0345 | 0.0086 | 0.2053 | 0.1336 | 0.0374 | 0.0196 | 0.5322 | 0.2982 |

### E.4 EVALUATION METRICS

We use RMSE (Root Mean Square Error) and MAE (Mean Absolute Error) as the primary metrics for comparison with baseline models. These metrics are commonly employed in regression tasks and effectively assess the disparity between predicted and actual values.

### E.5 ADDITIONAL EXPERIMENTAL RESULTS

In this section, we present additional experimental results that further validate the effectiveness of our proposed AhaTrans model. We first compare AhaTrans with conventional spatio-temporal prediction methods under various data conditions, and then extend our comparison to include state-of-the-art foundation models and specialized time series methods.

#### E.5.1 PERFORMANCE COMPARISON WITH TRADITIONAL BASELINES

Due to page limitations in the main text, we provide here a more comprehensive analysis comparing AhaTrans with various traditional prediction methods, including MetaST, RegionTrans, STResNet, ConvLSTM, and ARIMA, across different datasets. As shown in Table 5, AhaTrans consistently outperforms these methods across all testing scenarios. Under various data-scarce conditions (7 days, 15 days, and 30 days), AhaTrans demonstrates the lowest Root Mean Square Error (RMSE) and Mean Absolute Error (MAE), confirming its high reliability across all data scales. These results further validate the superior performance of our proposed approach and its robust adaptability to different prediction tasks.

#### E.5.2 COMPARISON WITH FOUNDATION MODELS AND ADVANCED TIME SERIES METHODS

Given recent advances in spatiotemporal data foundation models, we expand our comparative analysis to include state-of-the-art methods in this domain. While our primary benchmarking study focuses on transfer learning approaches that align with the AhaTrans problem formulation, we also recognize the importance of comparative evaluation against broader methodological paradigms.

Regarding data configuration, we employ the NYCBike and DCBike datasets as pretraining data, with each dataset comprising one complete year of spatiotemporal observations (2,242,560 spatiotemporal data points), providing a comprehensive foundation for cross-city spatiotemporal pattern learning. CHIBike and DCTaxi serve as target domains for evaluation, with data preprocessing following identical spatiotemporal gridding strategies. All models utilize one month of target city data for fine-tuning, while the remaining 11 months are reserved for performance evaluation, ensuring comprehensive validation of model generalization capabilities. Following baseline approaches, we preserve the original numerical ranges of all datasets without standardization or normalization to maintain consistency with original data distributions and prevent information loss. This design choice

Table 6: Performance comparison with foundation models and time series methods

| Method | CHIBike | | DCTaxi | |
|---|---|---|---|---|
| | RMSE | MAE | RMSE | MAE |
| PatchTST | 14.29 | 6.57 | 19.58 | 8.97 |
| UrbanGPT | 6.92 | 4.63 | 9.37 | 5.91 |
| UniST | 6.84 | 3.95 | 7.81 | 5.27 |
| AhaTrans | **3.66** | **1.58** | **4.45** | **2.09** |

requires different methods to directly handle numerical variations and distributional characteristics inherent in the raw data, thereby more authentically reflecting real-world application challenges.

For baseline method configurations, PatchTST employs `patch_len=16` and `stride=8` settings, segmenting each time series into overlapping temporal patches, with 100 epochs of pretraining followed by 50 epochs of fine-tuning. This configuration effectively captures local temporal pattern features. UrbanGPT is implemented based on the GPT-2 base architecture, with `context_length` configured to 168 hours (one week) and `embedding_dim` set to 768, modeling long-term dependencies in spatiotemporal sequences in an autoregressive manner. UniST adopts a unified spatiotemporal representation learning framework with `spatial_dim` set to 64 and `temporal_dim` set to 128, capturing complex interactions in urban dynamic patterns through joint optimization of spatiotemporal embeddings.

The experimental results demonstrate that AhaTrans exhibits substantial performance superiority across both target domains. On the CHIBike dataset, AhaTrans achieves a 46.5% improvement in RMSE (from 6.84 to 3.66) and a 60.0% improvement in MAE (from 3.95 to 1.58) compared to the best-performing baseline method, UniST. When compared to the traditional time series approach PatchTST, AhaTrans delivers performance improvements exceeding 74%.

On the DCTaxi dataset, AhaTrans outperforms UniST with a 43.0% improvement in RMSE (from 7.81 to 4.45) and a 60.3% improvement in MAE (from 5.27 to 2.09). These results convincingly demonstrate AhaTrans' robust generalization capabilities in cross-modal transportation transfer scenarios.

Analyzing the performance differences across methods reveals distinct characteristics. PatchTST, as a purely temporal modeling approach, exhibits the poorest performance among all methods, with the highest RMSE and MAE values, primarily due to its lack of spatial correlation modeling capabilities when handling complex city-level spatiotemporal patterns. UrbanGPT, while possessing reasonable sequence modeling capabilities, suffers from overfitting risks in fine-grained spatiotemporal prediction tasks, and its large parameter space shows constrained adaptability when trained on limited fine-tuning data, resulting in performance intermediate between PatchTST and UniST. UniST demonstrates strong performance in unified spatiotemporal representation learning and ranks as the top-performing baseline method; however, it exhibits limitations in cross-city knowledge transfer, particularly lacking the adaptive alignment mechanisms that characterize AhaTrans.

Furthermore, AhaTrans's computational efficiency (only 5M parameters compared to UniST's and UrbanGPT's billion-scale models) makes it significantly more deployable in resource-constrained real-world traffic management systems that require responsive predictions.

# F    MORE MODEL ANALYSIS

## F.1    FINE-GRAINED ABLATION STUDY

### F.1.1    ABLATION STUDY ON THE STCE MODULE

In scenarios with limited data availability in target cities, precise feature learning becomes a critical factor for successful modeling. To verify this hypothesis, we conducted comprehensive ablation experiments across both temporal and spatial dimensions under the 7-day data availability condition, and evaluated performance using Root Mean Square Error (RMSE) metrics.

As illustrated in Table 7, STCE and its temporal and spatial contrastive sub-modules significantly reduced prediction errors under extreme data scarcity, thereby substantiating the essential role of STCE. Furthermore, our experiments revealed that the Spatial Contrastive Learning (SCL) module exhibited superior performance in capturing cross-city spatial similarities during cross-city prediction tasks. Conversely, for intra-city transfer scenarios where spatial features already exhibit inherent

Table 7: Ablation study on STCE module within AhaTrans, showing RMSE results with 7-day target city data. Analysis includes progressive removal of Temporal Contrastive Learning (TCL), Spatial Contrastive Learning (SCL), and the entire STCE module. The best results are marked in **bold**.

| Method | AhaTrans | w/o STCE | w/o SCL | w/o TCL |
|---|---|---|---|---|
| NYCBike $\rightarrow$ CHIBike | **0.0216** | 0.0225 | 0.0221 | 0.0219 |
| DCBike $\rightarrow$ NYCBike | **0.0379** | 0.0388 | 0.0385 | 0.0383 |
| NYCBike $\rightarrow$ DCBike | **0.0276** | 0.0309 | 0.0302 | 0.0284 |
| DCBike $\rightarrow$ DCTaxi | **0.0280** | 0.0313 | 0.0289 | 0.0306 |

similarities, the Temporal Contrastive Learning (TCL) module was found to be more effective in improving prediction accuracy.

### F.1.2 ABLATION STUDY ON THE GTEN MODULE

As the core module of AhaTrans, GTNE explicitly differentiates between shared experts and city-specific experts, facilitating effective knowledge transfer from source cities while filtering out non-transferable patterns. To investigate the mechanisms by which different expert components extract and process domain-specific knowledge, we conducted systematic ablation experiments by isolating the contributions of city-specific and shared experts, measuring performance using Mean Absolute Error (MAE). We evaluated three model configurations:

1. Complete AhaTrans: The full architecture, incorporating source city experts, target city experts, and shared experts

2. Without Shared Experts (w/o Shared): A variant where shared experts were removed, creating direct connections between source city experts and the target task layer

3. Without Shared and Target Experts (w/o Shared&Target): A simplified version using only source city experts, following a pre-training on source data and fine-tuning on target data paradigm

Table 8 demonstrates that across all transfer scenarios, the complete AhaTrans model consistently achieved superior performance with the lowest MAE values, validating the effectiveness of our proposed GTNE architecture. Our detailed analysis revealed:

- Contribution of Shared Experts: Comparative analysis between the complete model and the "w/o Shared" variant revealed performance degradation when shared experts were removed. For instance, in the NYCBike to CHIBike transfer scenario, MAE increased from 0.0059 to 0.0064. These results suggest that shared experts successfully capture generalizable knowledge patterns across cities while mitigating the transfer of domain-specific, potentially detrimental information.

- Importance of Target Experts: Comparison between the "w/o Shared" and "w/o Shared & Target" configurations demonstrated that further removal of target experts led to additional performance deterioration. In the DCBike to NYCBike transfer scenario, for example, MAE increased from 0.0131 to 0.0133. This finding underscores the critical function of target city experts in preserving and modeling city-specific spatio-temporal dynamics and mobility characteristics. These unique target domain features would typically be overshadowed or suppressed in single-expert architectures dominated by source domain knowledge. The inclusion of dedicated target experts ensures the model maintains target-specific representations while simultaneously leveraging transferable knowledge from source domains, resulting in more balanced and effective knowledge transfer.

These ablation study results provide strong empirical evidence that the combination of shared experts, which extract cross-city commonalities while isolating non-transferable information, and target city experts, which preserve city-specific spatio-temporal patterns, enables AhaTrans to achieve efficient and effective cross-city knowledge transfer for urban mobility prediction.

### F.2 GENERALIZABILITY ANALYSIS

### F.2.1 ADAPTABILITY ACROSS DIFFERENT GRID RESOLUTIONS

To comprehensively evaluate the adaptability and scalability of our approach across different spatial granularities, we conducted additional experiments using higher-resolution grid partitioning. Specifically, we extended our evaluation to the Beijing-to-Chengdu transfer task using $32 \times 32$

Table 8: Ablation study on GTEN module within AhaTrans, showing MAE results for cross-city transfer tasks. "w/o Shared" and "w/o Shared & Target" represent progressive removal of shared and target city experts respectively. The best results are marked in **bold**.

| Method | AhaTrans | w/o Shared | w/o Shared & Target |
|---|---|---|---|
| NYCBike → CHIBike | **0.0059** | 0.0064 | 0.0067 |
| DCBike → NYCBike | **0.0125** | 0.0131 | 0.0133 |
| NYCBike → DCBike | **0.0075** | 0.0085 | 0.0089 |
| DCBike → DCTaxi | **0.0056** | 0.0062 | 0.0064 |

grid configurations, compared to the $16 \times 16$ grids employed in other tasks. This evaluation serves multiple purposes: (1) it assesses whether our adaptive hypernetwork architecture can effectively handle increased spatial complexity and finer-grained urban patterns; (2) it validates the scalability of our approach when the number of spatial units increases from 256 to 1,024 regions; and (3) it demonstrates the robustness of cross-city knowledge transfer mechanisms under varying levels of spatial detail.

The experimental setup maintains consistency with our primary evaluation protocol. We utilize the complete Beijing taxi dataset (trajectories from November 2015 to April 2016) as the source domain for model training. For the target domain, we employ the Chengdu taxi dataset (November 2016) under data-scarce conditions with only 10 days of training data. This restrictive data availability scenario allows us to evaluate the practical utility of our transfer learning approach when both spatial complexity and data scarcity present challenges simultaneously.

Table 2 presents the performance comparison results under these higher-resolution configurations. The experimental results demonstrate that AhaTrans maintains superior performance even with increased grid resolution, achieving **4.95%** in RMSE and **10.16%** in MAE compared to the second-best baseline CrossTReS. These results validate several key aspects of our approach: (1) the adaptive hypernetwork architecture successfully scales to handle quadrupled spatial complexity without performance degradation; (2) the cross-city knowledge transfer mechanisms remain effective when modeling finer-grained spatiotemporal patterns; and (3) our method demonstrates consistent robustness across different levels of spatial granularity. This adaptability across varying grid resolutions confirms that AhaTrans can be flexibly deployed in diverse urban analytics scenarios with different spatial monitoring requirements and complexity constraints.

### F.2.2 ADAPTABILITY ACROSS DIFFERENT SPATIO-TEMPORAL SCENARIOS

It is important to emphasize that while our experiments have primarily focused on traffic flow prediction, the AhaTrans framework—with its STCE, GTEN, and TBR modules—is not specifically designed for traffic applications and can be generalized to other spatio-temporal prediction scenarios. Our choice of traffic flow prediction as the application domain is deliberate for two key reasons: (a) it serves as a representative task in spatio-temporal prediction with well-established benchmark datasets, and (b) as illustrated in Figure 1, it clearly demonstrates the shortcomings of existing methods in addressing challenges such as inaccurate feature extraction and negative transfer.

To empirically validate the generalizability of our approach, we conducted additional experiments on cross-city crime prediction—a distinct spatio-temporal prediction task. We utilized the Chicago crime dataset (Crimes - 2001 to Present) as the source domain and the New York City crime dataset (NYPD Complaint Data) as the target domain. For both cities, we partitioned the geographical area into a $32 \times 32$ grid and aggregated crime incidents in each grid cell at hourly intervals.

In our transfer learning setup, we pre-trained AhaTrans on the complete training set of the source city (Chicago) and simulated data scarcity in the target city (NYC) by utilizing only 10% of its available training data. This experimental configuration enabled us to rigorously evaluate the effectiveness of AhaTrans in data-scarce scenarios for a fundamentally different type of spatio-temporal prediction task, thereby demonstrating the framework's versatility beyond traffic-related applications.

Table 9 presents the performance comparison between AhaTrans and other transfer learning baselines on the crime prediction task. The results demonstrate that AhaTrans substantially outperforms all baseline methods, achieving a 53.1% reduction in RMSE and a 39.2% reduction in MAE compared to the next best performing method (CrossTReS).

These results confirm that AhaTrans can effectively generalize to diverse spatio-temporal prediction tasks beyond traffic flow prediction. The significant performance improvement in crime predic-

Table 9: Performance comparison on cross-city crime prediction from Chicago to NYC.

| Metric | MetaST | | CrossTReS | | TransGTR | | AhaTrans | |
|--------|--------|-----|-----------|-----|----------|-----|----------|-----|
| | RMSE | MAE | RMSE | MAE | RMSE | MAE | RMSE | MAE |
| Value | 14.98 | 6.85 | 8.87 | 2.83 | 9.51 | 3.96 | **4.16** | **1.72** |

Table 10: Performance comparison with different numbers of attention heads.

| Number of Heads | 4 | 8 | 16 |
|-----------------|-------|--------|--------|
| RMSE | 0.375 | 0.0369 | 0.0381 |
| MAE | 0.0122 | 0.0119 | 0.0124 |

tion highlights the versatility of our approach and suggests that the principles underlying Aha-Trans—particularly the combination of contrastive embedding, graph transformer encoding, and bidirectional regularization—provide a robust foundation for transfer learning across different urban spatio-temporal prediction challenges.

### F.3 EFFECT OF NUMBER OF ATTENTION HEADS

The selection of an appropriate number of attention heads ($h$) is constrained by the dimension of the feature space $d_{model}$. To ensure an equal division of features across heads, $h$ must be a factor of $d_{model}$. With our implementation using $d_{model} = 64$, the valid options for $h$ include $\{1, 2, 4, 8, 16, 32, 64\}$.

While increasing the number of heads can theoretically provide greater representational capacity and modeling diversity, excessively large values (e.g., $h = 64$, which results in $d_k = 1$) may lead to overly fragmented representations that harm model performance. Conversely, multiple attention heads generally enable greater parallelism and diverse feature modeling, which modern GPUs can efficiently exploit.

In AhaTrans, we set $h = 8$, providing sufficient expressiveness (with $d_k = 8$) to capture diverse spatiotemporal dependencies while maintaining high computational efficiency. To validate this design choice, we evaluated the impact of varying $h$ on model performance using the DCBike $\rightarrow$ NYCBike transfer task. As shown in Table 10, AhaTrans demonstrates robust performance across different values of $h$, with $h = 8$ yielding the optimal balance between representational capacity and computational efficiency.

The results indicate that while the model maintains relatively stable performance across different head configurations, $h = 8$ produces the best overall results with the lowest error metrics. This empirical finding supports our theoretical understanding that an intermediate number of heads provides an optimal balance between representational diversity and feature coherence.

### F.4 CASE STUDY ON SPATIAL PERSPECTIVE

In Figure 7 (right), we have visualized the temporal dimension of our analysis. Given that spatio-temporal data exhibits heterogeneous characteristics in both spatial and temporal dimensions, we conducted detailed case studies to demonstrate the efficacy of our proposed model. Figure 9 presents heatmaps comparing the traffic flow predictions generated by AhaTrans and STAN methods against ground truth data for the NYCBike dataset. We analyzed three representative periods on January 1, specifically morning (08:00-09:00), afternoon (14:00-15:00), and evening (21:00-22:00). These periods were selected to encompass both peak traffic hours and a low-volume nighttime interval. The figure reveals substantial variations in bicycle demand patterns across these periods. Bicycle usage frequency exhibited markedly higher values during peak hours and a significant decrease during the evening period. Our experimental results demonstrate that AhaTrans predictions aligned more closely with ground truth observations, as evidenced by a mean square error (MSE) of 2.216 compared to STAN's 3.992. This significant improvement in prediction accuracy substantiates the superior robustness and precision of the AhaTrans framework for spatio-temporal data prediction tasks.

### F.5 ANALYSIS OF MODEL PARAMETERS

We conducted a parameter analysis to ensure fair comparisons across methods, summarizing the parameter counts for each approach as shown in Table 11. Our analysis reveals that AhaTrans contains 4.73M parameters in total, which is 25.9% fewer than STAN (6.38M) while achieving superior traffic

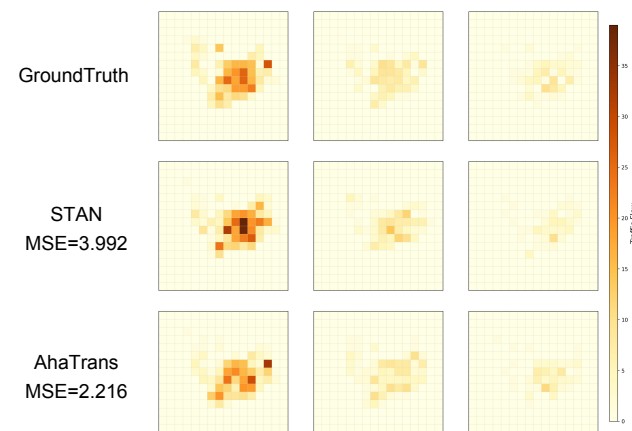

Figure 9: The case study of the prediction results of STAN and AhaTrans in three time intervals 8:00-9:00am (left), 14:00-15:00pm (middle), and 21:00-22:00pm (right).

prediction accuracy. This demonstrates the efficiency of our model design, which effectively balances parameter count and prediction performance.

When compared to CrossTRes (Jin et al., 2022), we maintained the same feature extraction network (1.26M parameters) for fair comparison. While AhaTrans has a larger prediction network (3.48M vs 0.34M), it's important to note that the original CrossTRes requires additional computational overhead during training due to its Domain Adaptation components (including Node-level and Edge-level Adaptation). These adaptation mechanisms, though not reflected in the parameter count, significantly increase training time and resource requirements.

Therefore, although AhaTrans has a moderately higher parameter count than CrossTRes, it delivers substantially better prediction performance without the computational burden of domain adaptation procedures, resulting in better overall efficiency. This highlights AhaTrans's ability to effectively leverage its parameters for improved prediction accuracy while maintaining computational efficiency.

Table 11: Comparison of model parameters across different methods

| Method | Feature Network (M) | Prediction Network (M) | Total (M) | Rel. to STAN (%) |
|---|---|---|---|---|
| STAN | 6.11 | 0.27 | 6.38 | 100.0 |
| CrossTRes | 1.26 | 0.34 | 1.60 | 25.1 |
| AhaTrans | 1.26 | 3.48 | 4.73 | 74.1 |

# G   LIMITATIONS AND FUTURE WORK

## G.1   LIMITATIONS

Although AhaTrans exhibits promising performance in cross-city traffic flow prediction tasks, it continues to encounter fundamental challenges inherent to transfer learning methodologies, particularly regarding source city selection strategies. A significant unresolved issue remains the identification and optimal matching of source cities that offer maximum transfer utility when confronted with a novel target city. The current implementation of AhaTrans depends primarily on manual source city designation, without incorporating an automated, data-driven matching framework. This limitation substantially constrains its scalability and generalization capabilities when deployed across extensive multi-city environments.

## G.2   FUTURE WORK

Our future research will prioritize several critical areas to enhance the effectiveness and broader applicability of our framework:

- **Expansion to Diverse Data Types.** In our ongoing efforts, we will strive to extend the framework's capabilities to handle a wider range of data types, including satellite imagery, IoT sensor data, and social media feeds. This expansion will enable a more comprehensive understanding of urban dynamics, integrating multiple data sources to enhance urban planning, management, and decision-making processes. The ability to process these diverse data types will provide a richer,

multidimensional view of cities, facilitating more informed and strategic urban development initiatives.

- **Enhancement of Feature Extraction Networks.** We will focus on advancing the framework's ability to capture and process complex urban data more effectively. This includes refining the feature extraction networks to enhance the representation of intricate urban dynamics. Enhanced networks will extract higher-level features, providing more meaningful insights into urban systems and enabling more accurate predictions.

- **Automated Source City Selection Mechanism.** To overcome the limitations of manual source city specification in current transfer learning frameworks, we plan to develop an automated mechanism for source city selection. By integrating multi-dimensional similarity features—such as traffic flow patterns, spatial structure, infrastructure layout, and socio-economic attributes—between cities, we aim to construct a transferability metric model that quantifies the suitability of source-target city pairs. Furthermore, we will explore graph-based modeling and meta-learning techniques to enable adaptive selection of optimal source cities from multiple candidates, thereby enhancing the generalization capability and deployment scalability of the framework in large-scale cross-city scenarios.

- **Advancement of Cross-Domain Transfer Learning Techniques.** The development of advanced transfer learning methods will be prioritized, emphasizing improved model adaptability across diverse urban contexts. These strategies will focus on minimizing domain shifts and enhancing performance when transferring models across different geographical locations and urban domains.

- **Development of Lightweight Training Models.** A key area for improvement will be the creation of more efficient models that require fewer computational resources. By optimizing performance and reducing complexity, we ensure the framework can be used across urban centers with varying capabilities.

