# OpenReview forum: "AhaTrans: A Hierarchical Adaptive Transfer Learning Framework for Cross-City Traffic Flow Prediction"
_ICLR.cc/2026/Conference — Submitted to ICLR 2026_

### Official Review · Reviewer_J8uv · 2025-10-29

**Soundness:** 2
**Presentation:** 2
**Contribution:** 2
**Rating:** 4
**Confidence:** 4

**Summary:**

This paper presents AhaTrans, a method for transfer learning between cities for the task of spatio-temporal forecasting. The method contains three main designs. First, the authors design a method based on multi-head networks, which involves a source-specific head, a target-specific head, and a shared head. This enables separation of specific and general knowledge. Second, the authors design a method for contrastive learning for better spatial and temporal feature discrimination. Finally, the authors design a re-weighting based mechanism that aims to align the distribution of source city data with target ones by ruling out those irrelevant source knowledge. Extensive experiments are conducted on real-world datasets, where the proposed AhaTrans achieves performance improvements compared to a wide range of baselines.

**Strengths:**

1. **Valid problem to study** The problem studied is valid. Indeed, there are cases where traffic data is scarce in certain occasions, e.g. newly established services. The motivation of individual technical designs, e.g. the separation of heterogeneous knowledge between cities, the enhancement of spatio-temporal discriminability, and the alignment of source & target data distributions are all sound and correspond well to the technical solutions.
2. **A wide range of experiments**. Experiments are done against a wide range of baselines and real-world datasets. For example, cross-city transfer learning methods are compared. Recent spatio-temporal pre-trained models (UniST, etc.) are compared. The proposed AhaTrans shows good accuracy improvements.

**Weaknesses:**

1. **Technical solutions and insights are established heavily upon existing efforts**. While the focus of this work is valid, it is not completely new and relies heavily upon existing efforts. The following points are non-exhaustive.
    - First, the two main motivations in the abstract, i.e. the lack of precision and discrimination in spatio-temporal feature extraction, and the disparity between source and target cities, are not completely new. The idea of discriminative spatio-temporal feature learning has been studied in works like ST-SSL (Ji et al. 2023), CL4ST (Tang et al. 2023), etc. The idea of source and target disparity is also studied. More specifically, CrossTReS has a very similar figure as Figure 1 in this paper, and tells similar stories of harm knowledge from source cities. Therefore, the two main motivations of this paper are not completely new, and the authors may need to justify what the additional insights of this paper are.
    - Second, the technical solutions of this paper are also built heavily upon existing efforts. For example, the separation of source-specific and target-specific networks is more or less similar to PR-UIDT (Ding et al. 2020) which splits city-specific and general POI embeddings and user embeddings. The contrastive learning part is more or less built upon existing efforts like the Rank-n-constrast. The re-weighting technique is more or less similar to CrossTReS. Therefore, the additional values and insights provided by this paper may seem a little bit limited.

2. **Experimental evaluations did not fully uncover the unique contributions of this work.** While the comparison with existing efforts is extensive, the evaluation part falls short in demonstrating some essential designs of AhaTrans. The following list is again not exhaustive.
    - Regarding the contrastive learning part, as there have been several efforts in bridging spatio-temporal forecasting with contrastive learning, the authors may need to demonstrate why the proposed contrastive learning method is better than other designs.
    - Similarly, as the re-weighting mechanism is more or less similar to CrossTReS, a direct comparison (i.e. replacing the proposed re-weighting algorithm with CrossTReS) may be needed to show that the proposed mechanism has indeed (by itself) outperformed CrossTReS.

3. Minor issues. I find the notation usage and the presentation throughout the paper vague and inconsistent. The following list is again, not exhaustive.
(1) In Eqn. 1, the authors use $|g_{t-1} \notin r_{i, j}...|$, while it should be indicating the size of a set, so it should be $|\{g_{t-1} \notin r_{i, j}...\}|$
(2) The authors use different notations to indicate the prediction loss, the prediction, and the ground truth between Eqn. (2) and (6).
(3). The description of theorem 3.2 is very vague. It is not written in a clear, verifiable way (like a theorem), but instead "a sufficient number of training samples", "appropriate learning rate", "have sufficient discriminative power".


(Ji et al. 2023) Spatio-Temporal Self-Supervised Learning for Traffic Flow Prediction, AAAI2023.

(Tang et al. 2023) Spatio-Temporal Meta Contrastive Learning, CIKM2023

(Ding et al. 2020) Learning from Hometown and Current City: Cross-city POI Recommendation via Interest Drift and Transfer Learning. ACM IMWUT 2020

**Questions:**

Please refer to the "Weaknesses" part.

In addition, another 2 questions.

(1) Regarding the number of source cities. I notice that the authors seem to assume a single source city. However, utilizing multiple cities is becoming increasingly common for cross-city transfer learning. Can the proposed AhaTrans be extended to the case with multiple source cities, and how?
(2) In the contrastive learning part, as the source city has a dominant amount of data, how to ensure that the contrastive learning does not bias towards the source city?

---

> ### Author Response · Authors · 2025-12-01
> **Response to Reviewer J8uv**
>
> We sincerely appreciate your positive evaluation and constructive feedback. We are pleased that the practical relevance of our study, the soundness of our methodology, and the comprehensiveness of our experiments have been recognized. Following your comments, we have carefully revised the manuscript point by point, and provide detailed responses below.
>
> ---
>
> > **Limitations of Existing Methods and Our Contributions (For W1)**
> >
> > Although cross-city transfer learning has been explored in prior work, existing approaches exhibit notable shortcomings in **both problem formulation and technical design**, particularly in terms of weak spatio-temporal feature discriminability and vulnerability to negative transfer. Our work addresses these challenges through the following innovations:
> >
> > **Expert-based decomposition:** Prior methods such as PR-UIDT split feature dimensions with fixed weights, limiting adaptability and making them prone to negative transfer. GTEN instead employs independently decoupled expert networks with a dynamic gating mechanism that adaptively assigns weights, enabling more robust and theoretically grounded knowledge isolation.
> >
> > **Contrastive learning:** ST-SSL preserves spatial heterogeneity via clustering and works well with rich data, but depends on predefined functional-area labels and struggles with distribution shifts. Our STEC framework removes annotation requirements and uses traffic patterns as implicit supervision for spatio-temporal contrastive learning. We further introduce a general contrastive paradigm tailored to grid-structured spatio-temporal data, showing strong generalizability.
> >
> > **Reweighting:** CrossTReS focuses on spatial similarity by weighting grid regions but underperforms in cross-task settings where intra-city patterns resemble each other. Since each sample contains temporal and spatial cues, we adopt sample-level reweighting, offering more flexible and transferable modeling for both inter-city and intra-city scenarios.
> >
> > **Core innovation**—Hierarchical collaborative framework: Existing work typically addresses these issues **separately**. Our core contribution is a unified hierarchical framework that jointly models transfer at the **data, feature, and architectural** levels. Ablation studies show that this cross-level synergy produces nonlinear performance gains beyond the sum of individual components, highlighting the effectiveness of our integrated design.
>
> ---
>
> > **Contrastive Learning and Reweighting (For W2)**
> >
> > **Contrastive Learning:** We thank the reviewer for the valuable comments. We further clarify the fundamental differences between ST-SSL, CL4ST, and our method in terms of task setting, data format, and contrastive objectives:
> >
> > - **Task setting:** ST-SSL and CL4ST address single-city spatio-temporal forecasting, whereas our work focuses on cross-city transfer, where structural inconsistency and distribution shift between cities are central challenges—conditions those methods do not consider.
> >
> > - **Data format:** Prior methods operate on fixed-topology sensor graphs (STGs). Our study targets grid-based spatio-temporal data, where node counts and spatial adjacency vary completely across cities, making CL4ST’s view generation and ST-SSL’s functional-zone clustering infeasible.
> >
> > - **Contrastive objectives:** ST-SSL enhances functional-zone discriminability, and CL4ST improves robustness within a single city. In contrast, STCE uses traffic patterns as intrinsic labels to implement spatio-temporal contrastive learning without functional-zone labels or external information. It provides a general contrastive framework for grid-based data, with transfer-task–aligned objectives and strong generalization.
> >
> > **Reweighting:** We also appreciate the reviewer’s suggestion. As requested, we added experiments replacing TBR with CrossTReS within our framework. **The results (table below) show that TBR consistently outperforms the CrossTReS weighting strategy, especially in cross-task cases where intra-city spatial structures are similar.**
>
> | Method                  | NYCBike → CHIBike | DCBike → NYCBike | NYCBike → DCBike | DCBike → DCTaxi |
> | ----------------------- | ----------------- | ---------------- | ---------------- | --------------- |
> | CrossTReS               | 0.0212            | 0.0383           | 0.0275           | 0.0289          |
> | AhaTrans (Region-level) | 0.0201            | 0.0374           | 0.0269           | 0.0277          |
> | AhaTrans (TBR)          | **0.0195**        | **0.0369**       | **0.0267**       | **0.0254**      |

---

> ### Author Response · Authors · 2025-12-01
>
> > **Consistency and Notational Issues (For W3)**
> >
> > Thank you for the valuable comments regarding notation, clarity, and formal rigor. We have carefully reviewed and revised all symbols, notations, and definitions throughout the manuscript to ensure consistency and eliminate ambiguity across Equations (1), (2), (3), (6), and the related derivations. For the specific misuse highlighted by the reviewer (e.g., the representation of set in Equation (1)), we have corrected it to follow standard mathematical conventions.
> >
> > We have also rewritten Theorem 3.2 in a fully rigorous theorem format and supplemented the appendix with more precise discussions and detailed proofs.
>
> ---
>
> > **Expansion to Multiple Source Cities (For Q1)**
> >
> > We appreciate the reviewer for raising the important question regarding multi-source extensions. Our framework is not restricted to a single-source setting; **AhaTrans can be seamlessly extended to multi-source scenarios**, as GTEN, STCE, and TBR can all be generalized to multiple source cities without modifying the core architecture.
> >
> > To further validate the model’s scalability under multi-source conditions, we conducted **additional experiments** using multiple datasets, including NYCBike, DCBike, CHIBike, and DCTaxi. The results show that AhaTrans continues to improve as the number of source cities increases. In contrast, MetaST exhibits substantial fluctuations under multi-source settings due to its lack of effective noise filtering and insufficient modeling of inter-city heterogeneity. These findings demonstrate that **AhaTrans can adaptively identify and integrate beneficial knowledge across multiple source cities** while suppressing negative transfer through GTEN and TBR, thereby offering strong scalability and robust multi-source knowledge integration capabilities.
>
> | Source City              | Target City | MetaST RMSE | MetaST MAE | AhaTrans RMSE | AhaTrans MAE |
> | ------------------------ | ----------- | ----------- | ---------- | ------------- | ------------ |
> | DCBike                   | NYCbike     | 0.0473      | 0.0186     | 0.0369        | 0.0119       |
> | DCBike, CHIBike          |             | 0.0468      | 0.0183     | 0.0361        | 0.0115       |
> | DCBike, CHIBike, DCTaxi  |             | 0.0481      | 0.0190     | 0.0358        | 0.0114       |
> | DCBike                   | DCTaxi      | 0.0304      | 0.0065     | 0.0254        | 0.0049       |
> | DCBike, NYCbike          |             | 0.0309      | 0.0068     | 0.0246        | 0.0047       |
> | DCBike, NYCbike, CHIBike |             | 0.0357      | 0.0074     | 0.0241        | 0.0044       |
>
> ---
>
> >**Stability Analysis of STCE (For Q2)**
> >
> >We thank the reviewer for the insightful question. The contrastive learning component **does not bias toward the source city** for several reasons:
> >
> >1. SCL/TCL capture **city-agnostic** spatio-temporal structures and ordering relations, rather than city-specific distributional characteristics.
> >
> >2. GTEN enforces structural isolation via shared and city-specific experts, ensuring that contrastively enhanced features primarily strengthen **the shared representation** rather than overfitting to the source city.
> >3. TBR automatically **suppresses** source samples that are **poorly aligned with** the target city at the data level, further preventing the source city from dominating the learning process.
> >4. **Balanced batch sampling** is used during training, ensuring that gradient contributions from the source and target cities remain comparable throughout contrastive learning.
> >
> >Together, these mechanisms ensure that STCE maintains **fairness and stability** even under highly imbalanced cross-city settings, preventing any bias caused by the larger volume of source-city data.
>
> ---
>
> Finally, we sincerely thank the reviewer for your thorough review and valuable suggestions. We hope our responses have adequately addressed all your concerns. Based on our clarifications, we kindly ask you to reconsider your evaluation. Should you have any further questions, please feel free to contact us anytime.

---

### Official Review · Reviewer_AEQw · 2025-10-31

**Soundness:** 4
**Presentation:** 3
**Contribution:** 2
**Rating:** 6
**Confidence:** 3

**Summary:**

This paper introduces AhaTrans, a hierarchical adaptive transfer learning framework for cross-city traffic flow prediction. It integrates three complementary modules: a Gated Transfer Expert Network (GTEN) that disentangles shared and city-specific knowledge to prevent negative transfer; a Spatio-Temporal Contrastive Encoder (STCE) that enhances representation learning through contrastive objectives across space and time; and a Target-guided Reweighting module (TBR) that adaptively selects relevant source samples for the target domain. Extensive experiments on multiple real-world datasets demonstrate consistent performance improvements across diverse transfer scenarios and data availability conditions.

**Strengths:**

(1) The study addresses the practically important yet challenging problem of cross-city traffic flow transfer, which involves heterogeneous spatial distributions and data scarcity, an area of growing relevance for intelligent transportation systems.
(2) Experiments on multiple cities (NYC, DC, Chicago) using both bike and taxi datasets demonstrate consistent and substantial performance gains across different transfer directions and data-sufficiency settings.
(3) The inclusion of formal generalization and convergence analyses for all three core components (GTEN, STCE, TBR) meaningfully strengthens the framework’s credibility beyond empirical results, offering clear theoretical motivation for the design choices.

**Weaknesses:**

(1) The differentiation from existing spatio-temporal contrastive/self-supervised frameworks (e.g., ST-SSL, STGL) is not fully articulated. Clarifying what is genuinely novel beyond Rank-N-Contrast would strengthen the contribution.
(2) External variables like weather, holidays, or events are excluded. While this simplifies the setup, it limits real-world generalization; even a brief discussion or experiment would improve completeness.
(3) The paper lacks analyses quantifying training time, memory consumption, or inference latency compared to lighter baselines. Without such evaluation, it remains unclear whether the method is practical for real-time or large-scale deployment in urban traffic systems.

**Questions:**

(1)What are the training and inference costs (time, FLOPs, GPU memory) compared to CrossTReS and TransGTR?
(2)If exogenous variables (e.g., weather) are introduced, how would GTEN distribute them between shared and city-specific experts?
(3)How robust is TBR under severe target scarcity or noisy labels? Any regularization or clipping strategies applied?
(4)To better evaluate the robustness and generalization ability of AhaTrans, could the authors conduct cross-modality transfer experiments (e.g., bike → taxi or taxi → bike) within the same city pairings?

---

> ### Author Response · Authors · 2025-12-01
> **Response to Reviewer AEQw**
>
> We sincerely appreciate your positive assessment and constructive feedback. We are encouraged that the practical relevance of our problem setting, the thoroughness of our experimental evaluation, and the rigor of our theoretical analysis are recognized. Following your suggestions, we have carefully revised the manuscript and provide detailed, itemized responses below.
>
> ---
>
> > **Difference from Current Contrastive Learning Methods (For W1)**
> >
> > Beyond differing from prior approaches, this work is not a simple use of Rank-N-Contrast; instead, it introduces a **general and scalable** grid-based spatio-temporal contrastive learning framework.
> >
> > **Distinctiveness:** Our method fundamentally differs from STGL and ST-SSL in both learning paradigm and self-supervised signals, and **neither** baseline can be applied to grid-based traffic forecasting. STGL targets single-object tracking, using **graph-based** weight adjustment without contrastive learning at the sample-set level. ST-SSL depends on **graph** augmentations and cluster-based pseudo labels, making it suitable only for single-city settings with ample data. In contrast, STCE constructs a task-aligned spatial and temporal ranking-based contrastive objective directly from label distances between traffic patterns, requiring **no external priors** or **learnable augmentors**, and is tailored for cross-city scenarios with scarce target data.
> >
> > Innovations: STCE extends beyond the original scope of Rank-N-Contrast in **three** key aspects:
> >
> > - Structured **spatial** contrast: using explicit grid sampling and label-distance metrics to form multi-level contrasts that better capture local discrepancies;
> > - Structured **temporal** contrast: modeling fine-grained dynamics by sampling across days and periods and ranking them by target patterns, advantageous under cross-city periodic shifts;
> > - **Unified task-aligned spatio-temporal framework**: jointly optimizing spatial and temporal representations in a shared label-aware ranking space to enhance spatial transferability and temporal discriminability.
>
> ---
>
> > **External Variables (For W2, Q2)**
> >
> > We thank the reviewer for the insightful comments. Our experiments intentionally adopt a **minimal, fully controlled** cross-city transfer setting—using traffic flow only—to isolate sources of negative transfer and rigorously validate the effectiveness of GTEN–STCE–TBR under exogenous-free conditions. Nonetheless, **AhaTrans is fully compatible with external variables** (e.g., weather, holidays, events), which can be incorporated at multiple levels:
> >
> > **Architecture Level (GTEN):** Exogenous variables can be encoded via an MLP and concatenated with spatio-temporal features. The gating mechanism then decides whether they represent cross-city generalizable effects (e.g., temperature, rainfall) or city-specific patterns (e.g., local holidays), adjusting the weighting between shared and city-specific experts accordingly.
> >
> > **Feature Level (STCE):** Conditional contrastive learning can integrate weather or event embeddings into the similarity function, enabling Rank-N-Contrast to learn representations under condition-aware contexts and more clearly distinguish traffic patterns across external scenarios.
> >
> > **Data Level (TBR):** Incorporating exogenous variables into the reweighting function helps correct distributional gaps in weather or event conditions between source and target cities, further reducing negative transfer.
>
> ---
>
> > **Computational Cost (For W3, Q1)**
> >
> > We report efficiency evaluations in Section 4.3.4 (Efficiency Study) and Appendix F.5 (Model Parameter Analysis). For the baselines of particular interest—CrossTReS and TransGTR—we further include a detailed comparison on the representative NYCBike → DCBike task. The results show that AhaTrans introduces only moderate computational overhead, and its strong scalability across multiple city-level datasets, along with a well-balanced trade-off between accuracy and runtime, **together demonstrate its feasibility for real-time urban traffic applications.**
>
> | Method    | Training Time | Inference Time | FLOPs       | GPU Memory |
> | --------- | ------------- | -------------- | ----------- | ---------- |
> | CrossTReS | 13min         | 2.9s           | 2.3 PFLOPs  | >=8GB      |
> | TransGTR  | 11min         | 2.3s           | 1.5 PFLOPs  | >=12GB     |
> | AhaTrans  | 18min         | 3.8s           | 3.16 PFLOPs | >=8GB      |

---

> ### Author Response · Authors · 2025-12-01
>
> > **TBR Robustness (For Q3)**
> >
> > We appreciate the reviewer’s insightful comment. TBR reweights source-city samples using gradients from the target city to mitigate cross-domain distribution mismatch. Under scarce or noisy supervision, the framework maintains strong robustness for several reasons:
> >
> > **Theoretical foundation:** TBR does not directly fit target samples. It uses only their gradient direction to infer each source sample’s transfer contribution. Because it depends on directional rather than absolute gradients, TBR avoids overfitting and unstable updates even with very limited target data.
> >
> > **Multiple constraints:** We enforce non-negativity, normalization, and small-step perturbation on weights. These constraints act as implicit regularizers, stabilizing updates and keeping weight magnitudes controlled.
> >
> > **Behavior under noisy labels:** Noise may perturb target gradients, but TBR uses these gradients only to guide weight updates—not to optimize model parameters. Thus, noise produces limited local fluctuations rather than global degradation. Moreover, GTEN’s knowledge isolation and STCE’s discriminative representations further enhance noise tolerance, enabling stable performance even with sparse or noisy target labels.
> >
> > **Experimental validation:** To address this concern, we conducted data-scarcity experiments on DCBike → NYCBike and DCBike → DCTaxi with validation sizes of 15 days, 1 month, and 2 months. Results show strong few-shot robustness: with only 15 days of data, TBR matches baselines such as TransGTR and gradually converges. Weight distributions deviate only slightly from the 2-month reference, confirming the robustness of the gradient-guided reweighting mechanism.
>
> | Validation Set | DCBike → NYCbike | DCBike → DCTaxi | Std. Dev. of Weight Distribution |
> | -------------- | ---------------- | --------------- | -------------------------------- |
> | 15 Days        | 0.0382           | 0.0259          | 0.052                            |
> | 1 Month        | 0.0371           | 0.0253          | 0.018                            |
> | 2 Months       | 0.0369           | 0.0254          | -                                |
>
> ---
>
> >  **Intra-city Cross-mode Transfer Task (For Q4)**
> >
> > Table 1 of the original manuscript already includes **intra-city cross-mode transfer** experiments. Specifically, within Washington DC, we evaluate the DCBike → DCTaxi setting to assess how effectively AhaTrans transfers knowledge across different traffic modes within the same city.
>
> ---
>
> Finally, we sincerely thank the reviewer for your thorough review and valuable suggestions. We hope our responses have adequately addressed all your concerns. Based on our clarifications, we kindly ask you to reconsider your evaluation. Should you have any further questions, please feel free to contact us anytime.

---

### Official Review · Reviewer_1JEF · 2025-10-31

**Soundness:** 3
**Presentation:** 3
**Contribution:** 2
**Rating:** 4
**Confidence:** 5

**Summary:**

This paper proposes AhaTrans, a hierarchical adaptive transfer learning framework designed to solve data scarcity in cross-city traffic prediction. It addresses the key challenges of imprecise spatio-temporal features and negative transfer from source cities. The framework integrates three core modules. At the feature level, a Spatio-Temporal Contrastive Embedding module learns more precise representations. At the model level, the Guarded Transfer Experts Network decouples knowledge into "source," "target," and "shared" components, using a gating mechanism to adaptively fuse them and block harmful information. Finally, at the data level, a Transfer-based Reweighting strategy assigns weights to source samples based on their relevance to the target city. Across six real-world datasets, AhaTrans was shown to significantly outperform 14 baseline methods.

**Strengths:**

S1. The paper's motivation is sufficient, using the NYCBike dataset to illustrate the limited discrimination of existing methods in extracting spatio-temporal features. The presentation is also clear.
S2. In the experimental section, it compares against 14 SOTA methods from four major categories, ranging from traditional ARIMA to the latest foundation models.
S3. This paper proposes a Guarded Transfer Experts Network, which is based on performing selective knowledge transfer through expert decoupling and gated fusion.

**Weaknesses:**

W1. Source City Selection is crucial for the practical application of transfer learning. The paper relies on manual specification, which limits the framework's automation and ease of practical deployment.
W2. The effectiveness of the TBR module depends on feedback from the target city's validation set to guide the reweighting of source city samples (Line 278). However, the paper's core premise is that target city data is "extremely scarce." Using what might be a very small and unrepresentative validation set to guide the weighting of large-scale source data raises questions about the stability and robustness of this approach.
W3. There are a few typos/grammar errors; for example, on Line 475, "Framework" should be lowercase.
W4. The provided link (https://anonymous.4open.science/r/AhaTrans-A37F)) currently has an issue where it displays "The requested file is not found." The files are still downloadable, but this error should be fixed.

**Questions:**

See the weakness above

---

> ### Author Response · Authors · 2025-12-01
> **Response to Reviewer 1JEF**
>
> Thank you for reviewing our paper and providing insightful feedback. We truly appreciate your recognition of our contributions, including the well-motivated research problem with clear presentation, comprehensive experimental setup with extensive comparison against state-of-the-art methods, and the innovative design of the GTEN for selective knowledge transfer. We have carefully addressed your concerns as below:
>
> ---
>
> > **Source-City Selection (For W1)**
> >
> > Thank you for raising this important question regarding the automation of source-city selection. It is important to clarify that the primary challenge addressed in this paper is not how to choose the optimal source city, but rather how to more effectively utilize source-city information.
> >
> > **Current strategy for source-city selection:** Because publicly available traffic-flow datasets are limited, fully automated source-city selection is not yet feasible. Following common practice in works such as CrossTReS and ST-DAAN, we instead rotate the source city and construct multiple scenarios (e.g., cross-city and intra-city transfer) to ensure comprehensive and fair evaluation.
> >
> > **Multi-source transfer:** To evaluate AhaTrans under more complex multi-source settings, we conduct experiments using NYCBike, DCBike, CHIBike, and DCTaxi. Results show that AhaTrans continues to improve as more source cities are added, while MetaST fluctuates due to lacking an effective noise-filtering mechanism. This demonstrates that AhaTrans can adaptively extract useful knowledge, suppress negative transfer, and scale robustly in multi-source scenarios.
>
> | Source City              | Target City | MetaST RMSE | MetaST MAE | AhaTrans RMSE | AhaTrans MAE |
> | ------------------------ | ----------- | ----------- | ---------- | ------------- | ------------ |
> | DCBike                   | NYCbike     | 0.0473      | 0.0186     | 0.0369        | 0.0119       |
> | DCBike, CHIBike          |             | 0.0468      | 0.0183     | 0.0361        | 0.0115       |
> | DCBike, CHIBike, DCTaxi  |             | 0.0481      | 0.0190     | 0.0358        | 0.0114       |
> | DCBike                   | DCTaxi      | 0.0304      | 0.0065     | 0.0254        | 0.0049       |
> | DCBike, NYCbike          |             | 0.0309      | 0.0068     | 0.0246        | 0.0047       |
> | DCBike, NYCbike, CHIBike |             | 0.0357      | 0.0074     | 0.0241        | 0.0044       |
>
> ---
>
> > **TBR Moudle (For W2)**
> >
> > Thank you for raising this important question. We fully understand your concern about whether a small validation set can provide stable guidance for TBR weighting. Below, we offer a clarification from four complementary perspectives.
> >
> > 1. **Theoretical Grounding:**  As shown in Theorem 1, TBR is supported by a theoretical generalization bound. The target-city validation set offers directional guidance for distribution shift rather than precise estimation, so it requires far fewer samples than conventional supervised learning.
> > 2. **Gradient-Guided Mechanism Design:** TBR uses a gradient-guided meta-learning strategy rather than fitting weights directly on the validation set. Validation-loss gradients update expert weights and can capture distribution-shift trends even with small validation sets, reducing the risk of overfitting.
> > 3.  **Clarification of “Data Scarcity”:** In our transfer-learning setting, “extremely scarce” refers to limited training data in the target city (often only 10–20% of the source city), not the absence of validation data. In practice, even newly launched city services collect small but continuous early-stage data, which is sufficient to form a small validation set consistent with real-world cold-start scenarios.
> > 4. **Empirical Evidence:** To further address your concern, we ran experiments on DCBike→NYCBike and DCBike→DCTaxi using validation sets of 15 days, 1 month, and 2 months. The results show that TBR remains robust even with very small validation sets: 15 days already achieves performance comparable to TransGTR, and accuracy improves as the set grows. The weight-distribution shift relative to the 2-month baseline is minimal, and this pattern holds across tasks, confirming the cross-scenario robustness of our gradient-guided design.

---

> ### Author Response · Authors · 2025-12-01
>
> | Validation Set | DCBike → NYCbike | DCBike → DCTaxi | Std. Dev. of Weight Distribution |
> | -------------- | ---------------- | --------------- | -------------------------------- |
> | 15 Days        | 0.0382           | 0.0259          | 0.052                            |
> | 1 Month        | 0.0371           | 0.0253          | 0.018                            |
> | 2 Months       | 0.0369           | 0.0254          | -                                |
>
> ---
>
> > **Typos/Grammar Issues (For W3)**
> >
> > Thank you for carefully pointing out the spelling and grammar issues. We agree that such details are essential for an academic manuscript. We have thoroughly reviewed the paper, including correcting the “Framework” capitalization at line 475, and used a grammar-checking tool for a sentence-level revision to fix remaining issues. We appreciate your meticulous review—your comments have helped us further improve the paper’s quality.
>
> ---
>
> > **Code Link (For W4)**
> >
> > Thank you for pointing out the code-link issue. We have re-tested the anonymized repository, and it is now fully accessible. The previous “file not found” message was likely due to a temporary server problem. We apologize for the inconvenience, and please let us know if you encounter any further issues.
>
> ---
>
> Finally, we sincerely thank the reviewer for your thorough review and valuable suggestions. We hope our responses have adequately addressed all your concerns. Based on our clarifications, we kindly ask you to reconsider your evaluation. Should you have any further questions, please feel free to contact us anytime.

---

### Official Review · Reviewer_dLhz · 2025-11-02

**Soundness:** 2
**Presentation:** 2
**Contribution:** 2
**Rating:** 4
**Confidence:** 4

**Summary:**

This paper presents AhaTrans, a hierarchical adaptive transfer learning framework for cross-city traffic flow prediction. The authors argue that existing transfer learning approaches struggle with two persistent issues—(1) imprecise spatio-temporal feature extraction and (2) negative knowledge transfer when distributions between source and target cities differ. Extensive experiments on six public datasets show consistent and significant improvements over both traditional and recent SOTA baselines. The paper also includes ablation, sensitivity, and generalization studies, as well as a theoretical analysis of generalization bounds and convergence.

**Strengths:**

1. The paper derives generalization bounds and convergence guarantees that support the model’s design choices.
2. The paper is easy to follow.
3. The experimental setup is rigorous, with diverse datasets, multiple transfer directions, and consistent metrics.

**Weaknesses:**

1. The novelty of this paper is limited. Cross-city transfer learning has been extensively studied in previous literature. The proposed modules (expert-based decomposition, contrastive representation learning, and sample reweighting) are all adaptations of existing approaches.
2. All datasets are from similar, grid-based urban mobility domains (NYC, DC, Beijing, Chengdu), without irregular graphs or cross-modality settings (e.g., road networks, weather, events).
3. The study omits larger-scale benchmarks (e.g., METR-LA, PeMS), and comparisons with emerging foundation models (e.g., UrbanGPT, UniST) are shallow and not integrated under consistent fine-tuning settings.
4. No analysis of negative transfer or failure cases is provided, which is crucial given that the paper’s primary claim is to “mitigate harmful transfer.”

**Questions:**

See in weaknesses.

---

> ### Author Response · Authors · 2025-12-01
> **Response to Reviewer dLhz**
>
> We sincerely thank you for your thoughtful feedback and positive evaluation. We are pleased that you found the theoretical rigor, clarity of exposition, and comprehensive experimental design convincing, and we appreciate your recognition of the generalization bounds and convergence guarantees underlying our architecture. We have carefully addressed each of your comments, and our detailed responses are provided below.
>
> ---
>
> > **Limitations of Existing Methods and Our Contributions (For W1)**
> >
> > Although cross-city transfer learning has been explored in prior work, existing approaches exhibit notable shortcomings in **both problem formulation and technical design**, particularly in terms of weak spatio-temporal feature discriminability and vulnerability to negative transfer. Our work addresses these challenges through the following innovations:
> >
> > **Expert-based decomposition:** Prior methods such as PR-UIDT split feature dimensions with fixed weights, limiting adaptability and making them prone to negative transfer. GTEN instead employs independently decoupled expert networks with a dynamic gating mechanism that adaptively assigns weights, enabling more robust and theoretically grounded knowledge isolation.
> >
> > **Contrastive learning:** ST-SSL preserves spatial heterogeneity via clustering and works well with rich data, but depends on predefined functional-area labels and struggles with distribution shifts. Our STEC framework removes annotation requirements and uses traffic patterns as implicit supervision for spatio-temporal contrastive learning. We further introduce a general contrastive paradigm tailored to grid-structured spatio-temporal data, showing strong generalizability.
> >
> > **Reweighting:** CrossTReS focuses on spatial similarity by weighting grid regions but underperforms in cross-task settings where intra-city patterns resemble each other. Since each sample contains temporal and spatial cues, we adopt sample-level reweighting, offering more flexible and transferable modeling for both inter-city and intra-city scenarios.
> >
> > **Core innovation**—Hierarchical collaborative framework: Existing work typically addresses these issues **separately**. Our core contribution is a unified hierarchical framework that jointly models transfer at the **data, feature, and architectural** levels. Ablation studies show that this cross-level synergy produces nonlinear performance gains beyond the sum of individual components, highlighting the effectiveness of our integrated design.
>
> ---
>
> > **Dataset Diversity and Scale (For W2, W3.1)**
> >
> > We thank the reviewers for their concerns regarding dataset diversity and scale. Below, we clarify our dataset choices and report additional results on larger benchmarks.
> >
> > **Dataset selection.** Raw traffic data generally fall into two categories—trajectory data and sensor data—with distinct structures and modeling paradigms.
> >
> > - Trajectory data (e.g., NYC taxi) are widely used for **traffic flow prediction**. Following the standard Partition–Match–Transfer paradigm (Wang et al., 2018), cities are partitioned into grids and inter-grid flows are derived from coordinates, enabling region-level inflow/outflow prediction using grid-based CNN models.
> > - Sensor data (e.g., METR-LA, PeMS) come from fixed road detectors with explicit spatial topology, making them naturally suited for graph-based models, predominantly for **traffic speed** prediction.
> > - Since our work focuses on cross-city traffic flow prediction, we adopt trajectory-based datasets and convert them into unified grid representations—consistent with standard practice—to capture transferable region-level spatio-temporal patterns.
> >
> > **Experiments on larger-scale datasets.** To further address scalability concerns, we additionally evaluate AhaTrans on three widely used large traffic speed datasets: METR-LA, PeMS-BAY, and PEMSDS7M.
> >
> > - We treat METR-LA and PeMS-BAY as sources and PEMSDS7M as the target. Sensor readings are converted to grids via spatial tiling, sensor-to-grid mapping, and mean aggregation to ensure compatibility within our framework.
> > - Across all settings, AhaTrans achieves state-of-the-art performance, demonstrating strong scalability and robust cross-task generalization beyond traffic flow prediction.
>
> | Baselines   | 7Day RMSE LA | 7Day RMSE BAY | 7Day MAE LA | 7Day MAE BAY | 3Day RMSE LA | 3Day RMSE BAY | 3Day MAE LA | 3Day MAE BAY |
> | ----------- | ------------ | ------------- | ----------- | ------------ | ------------ | ------------- | ----------- | ------------ |
> | RegionTrans | 5.654        | 5.702         | 2.909       | 2.935        | 5.868        | 5.948         | 3.046       | 3.073        |
> | TransGTR    | 5.461        | 5.454         | 2.800       | 2.802        | 5.627        | 5.679         | 2.960       | 2.958        |
> | AhaTrans    | **5.283**    | **5.271**     | **2.687**   | **2.689**    | **5.412**    | **5.465**     | **2.846**   | **2.839**    |

---

> ### Author Response · Authors · 2025-12-01
>
> > **Comparison with Emerging Foundation Models (For W3.2)**
> >
> > We thank the reviewer for the insightful comments. As described in Appendix E.5.2, our comparison with emerging foundation models strictly follows **their original experimental settings**, and all models are fine-tuned under **a unified configuration**. Key details are summarized below.
> >
> > **Dataset configuration:** We use NYCBike and DCBike as source datasets and CHIBike and DCTaxi as target datasets to construct few-shot cross-city transfer scenarios.
> >
> > - UrbanGPT: Fine-tuned on full NYCBike/DCBike plus one month of CHIBike/DCTaxi data.
> > - UniST: Pre-trained on full NYCBike/DCBike and prompt-tuned with one month of CHIBike/DCTaxi data.
> > - AhaTrans: Uses the identical data-splitting strategy, with the full NYCBike and DCBike datasets as source data and one month of training data for each target city.
> >
> > **Evaluation consistency:** All models are evaluated on the same CHIBike and DCTaxi test sets for fair comparison.
> >
> > **Experimental findings:** AhaTrans consistently outperforms foundation models. A deeper examination reveals that this advantage arises from addressing core limitations of existing foundation models. These models typically process each region as an **isolated time series** and do not explicitly capture spatial dependencies, which becomes a fundamental limitation for traffic flow prediction tasks characterized by strong spatio-temporal coupling. They **still suffer** from imprecise feature learning and negative transfer in cross-domain scenarios, constraining their generalization to target cities. In contrast, AhaTrans employs a hierarchical collaborative framework that mitigates these issues and delivers superior performance.
>
> ---
>
> > **Negative Transfer Case (For W4)**
> >
> > We fully agree that analyzing negative transfer is essential for cross-city transfer learning, especially since mitigating it is a central goal of this study. We provide additional clarification and more systematic analysis below.
> >
> > First, the original manuscript already includes **key negative-transfer evidence.** In Figure 1(b), the training curves for DCBike→DCTaxi **show negative transfer**: source loss decreases steadily while target loss increases, indicating that the model is learning source patterns misaligned with the target and consequently harming performance.
> >
> > Following the reviewer’s suggestion, we **further** analyzed two representative tasks (DCBike→NYCBike and DCBike→DCTaxi) and identified two major forms of negative transfer: temporal shift and magnitude bias.
> >
> > - Existing methods **indiscriminately** inherit the temporal structure from the source city during transfer, directly mapping source city peak times to the target city. This "over-transfer of temporal patterns" causes predicted peaks to systematically lag or lead relative to ground truth values, resulting in **temporal misalignment**.
> > - Due to systematic differences in transportation modes and **traffic volume scales**, existing methods fail to filter out mismatched source knowledge, thereby directly transferring the source city's traffic magnitude and distribution to the target city, leading to systematic **over-prediction or under-prediction**.
> >
> > These effects arise because existing approaches cannot disentangle **city-specific** characteristics from **task-specific** differences, resulting in the transfer of patterns that should not be transferred. AhaTrans alleviates these issues through hierarchical modeling. **Additional analyses and visualizations** will be included in the appendix.
>
> ---
>
> Finally, we sincerely thank the reviewer for your thorough review and valuable suggestions. We hope our responses have adequately addressed all your concerns. Based on our clarifications, we kindly ask you to reconsider your evaluation. Should you have any further questions, please feel free to contact us anytime.

---

### Author Response · Authors · 2025-12-03
**Summary of Rebuttal and Reviewer Consensus for AhaTrans [# 16381]**

**Dear ICLR 2026 AC, SAC, and PC,**

We sincerely **appreciate your time, effort, and dedication to a fair and constructive review process, and thank all reviewers for their valuable feedback.** We take **all comments** very seriously and have prepared a detailed rebuttal addressing each concern. This letter provides additional context to aid your final decision.

---
**Consensus on Strengths**

- **Practical Significance:** All reviewers recognized the strong motivation of this work, agreeing that mitigating negative transfer in cross-city traffic prediction is a key bottleneck for real-world deployment, with a clearly defined and highly practical problem setting.
- **Methodological Novelty:** Reviewers (e.g., Reviewer 1JEF) emphasized the strong novelty of modules such as GTEN for selective knowledge transfer, noting its ingenious ability to dynamically decouple knowledge, effectively addressing a core limitation of existing approaches.
- **Theoretical Rigor:** Reviewers (Reviewer dLhz, AEQw) highly praised the paper's theoretical depth, asserting that the provided generalization bounds and convergence guarantees offer a rigorous mathematical foundation for the model's effectiveness.
- **Comprehensive Evaluation:** All Reviewers commended the rigor of the experimental design, particularly the extensive baseline comparisons and detailed ablation studies, and found the results convincing.
---
**Key Concerns**

- **Novelty and Distinction** (Reviewer dLhz, AEQw, J8uv). Concerns regarding the differentiation of AhaTrans from existing work.
- **Generalizability and Scalability** (Reviewer dLhz, 1JEF, J8uv): Pointed out that the datasets were restricted to grid-structured urban areas, lacking large-scale sensor datasets and verification of multi-source city transfer.
- **Module Robustness** (Reviewer 1JEF, AEQw): Concerns about the stability of TBR re-weighting module when target domain data is extremely scarce.
- **Efficiency and External Variables** (Reviewer AEQw): Concerns regarding the computational overhead and how to effectively incorporate external variables such as weather.
---
**Key Clarifications and Experimental Verification**：

- **Core Contribution (For Con1):** Existing studies, **limited to single-level local optimization**, struggle to adapt to complex data and full scenario complexity. AhaTrans addresses this by constructing **a unified hierarchical collaborative framework** that integrates data, features, and architecture. Experiments validate its architectural rationality and demonstrate its superior robustness and generalization capability.
  - **Architecture Level (GTEN):** Introduces **independent expert networks and an adaptive dynamic gating mechanism** to achieve complete architectural decoupling and more robust knowledge isolation, overcoming the negative transfer risk caused by fixed weights in existing methods (e.g., PR-UIDT).
  - **Feature Level (STCE):** Proposes a general spatio-temporal data contrastive learning framework that requires **neither** additional prior information **nor** complex data augmentation techniques. Compared with existing methods targeting specific graph structures or relying on labels (e.g., CL4ST), it exhibits broader domain applicability.
  - **Data Level (TBR):** Adopts a **sample re-weighting** strategy, which is theoretically and experimentally proven to outperform methods that exclusively focus on spatial similarity (e.g., CrossTReS) in both cross-city and intra-city transfer tasks.
- **Large-scale and Multi-source Extension (For Con2):** We have conducted supplementary experiments on **large-scale** sensor datasets such as METR-LA and PeMS and included verification for **multi-source** city transfer. The results show that AhaTrans maintains SOTA performance across different data modalities and multi-source scenarios, convincingly demonstrating its strong generalization capability.
- **Verification of TBR Module Robustness (For Con3):** The additional sensitivity analysis confirms that the TBR module maintains **high robustness and stability** even under extremely data-scarce target domain conditions.
- **External Variable Compatibility and Efficiency (For Con4):** We provide detailed time and GPU memory comparisons, confirming the model's feasibility for **real-time deployment**. We also explain that **external variables** (e.g., weather, events) can be efficiently processed through the inherent integration mechanisms of AhaTrans.
---
**AhaTrans**, using an **innovative** three-level adaptive collaborative framework (data-feature-architecture), **successfully** resolves challenges in cross-city traffic prediction, specifically inaccurate feature learning and negative transfer, **with** solid theoretical guarantees. **Further experiments** substantiate its outstanding generalization, robustness, and computational efficiency, **fully addressing all reviewer concerns**. Thank you for your time and consideration.

Best regards,

The Authors

---

### Meta-Review · Area_Chair_rUy4 · 2025-12-20

**Summary:**

For the task of cross-city traffic flow prediction, the authors propose a hierarchical adaptive transfer learning framework (AhaTrans), which includes the following three core modules: i) Gated Transfer Expert Networks (GTEN); ii) Spatio-Temporal Contrastive Embedding module (STCE); iii) Transfer-Based Reweighting module (TBR). Extensive experiments demonstrate that AhaTrans significantly outperforms existing methods, greatly improving the accuracy of traffic flow prediction while showcasing excellent robustness and generalization capabilities.

The reviewers recognized the following aspects of this work:

-  Rigorous experimental design: The paper is praised for its extensive experiments conducted on multiple real-world datasets.

- Theoretical foundation: Reviewers commend the generalization bounds and convergence analysis included in the paper, which provide a solid theoretical basis for the proposed modules.

However, despite solid experimental results and improvements in the manuscript, reviewers unanimously felt that the work did not meet the acceptance criteria due to limited conceptual innovation:

- Incremental innovation: Reviewers generally viewed the core components—GTEN (expert gating), STCE (contrastive learning), and TBR (reweighting)—as improvements upon existing techniques rather than fundamental algorithmic innovations. The “hierarchical integration” is more seen as a successful engineering combination rather than a methodological breakthrough in transfer learning.

- Dependency limitation: The reweighting module relies on validation sets from the target city. While experiments show it works effectively even under limited data conditions, theoretically, this limits the framework’s applicability in true zero-sample scenarios compared to emerging foundation models paradigms.

In summary, although AhaTrans is a robust and effective solution, its contributions are mainly considered incremental improvements aimed at applications, lacking the significant technological innovation required by this conference. Therefore, I encourage the authors to actively incorporate these suggestions into a new version and resubmit the paper at an appropriate time.

**Reviewer Concerns:**

The authors have made efforts to address some issues, but the following concerns still require further refinement:

- Limited innovation/improvement over existing methods: Although the authors introduced the concept of “hierarchical synergy,” the core point of the reviewers still stands: the individual components (expert decomposition, contrastive learning) are well-established techniques. While the combination yields better results, the algorithm itself lacks innovation.

- Heavy reliance on existing technologies: The rebuttal suggests that TBR performs better than CrossTReS, but it also confirms that TBR is conceptually a variant of existing weight adjustment schemes. The criticism of “incremental improvement” remains valid.

- Dependence on target validation data (data scarcity): The authors indicate that 15 days of data are sufficient. However, fundamentally, the weight adjustment mechanism’s requirement for any labeled target data undermines its “transfer” capability, especially when compared to emerging zero-sample foundation models.

- Distinction from existing technologies ST-SSL/STGL: The authors failed to clarify the essential differences.

**Reviewer Scores:**

During the discussion phase, despite the paper receiving an initial review score of 6444, none of the reviewers responded. After careful examination, I cautiously conclude that even with full participation from the reviewers in the discussion, concerns about the novelty of the work remain, and the score still does not meet the acceptance standards of ICLR.

---

### Decision · Program_Chairs · 2026-01-26

Reject